

# Holocene land cover change in North America: continental trends, regional drivers, and implications for vegetation-atmosphere feedbacks

Andria Dawson*†[1], John W. Williams†[2], Marie-José Gaillard[3], Simon J Goring[2], Behnaz Pirzamanbein[4], Johan Lindstrom[5], R. Scott Anderson[6], Andrea Brunelle[7], David Foster[8], Konrad Gajewski[9], Daniel G. Gavin[10], Terri Lacourse[11], Thomas A Minckley[12], Wyatt Oswald[13], Bryan Shuman[12], Cathy Whitlock[14]

[1]Department of Mathematics and Computing, Department of Biology, Mount Royal University, Calgary, AB T3E6K6

[2]Department of Geography and Center for Climatic Research, University of Wisconsin-Madison, Madison, WI 53706

[3]Department of Biology and Environmental Science, Linnaeus University, SE 39231 Kalmar, Sweden

[4]Department of Statistics, School of Economics and Management, Lund University, SE-220 07 Lund, Sweden

[5]Division of Mathematical Statistics, Centre for Mathematical Sciences, Lund University, SE-221 00 Lund, Sweden

[6]School of Earth and Sustainability, Northern Arizona University, Flagstaff, AZ 86011

[7]Geography Department, University of Utah, Salt Lake City, UT 84112

[8]Harvard Forest, Harvard University, Petersham, MA 01366

[9]Département de Géographie, Environnement et Géomatique, Université d'Ottawa, Ottawa, ON K1N 6N50

[10]Department of Geography, University of Oregon, Eugene, OR 97403

[11]Department of Biology and Centre for Forest Biology, University of Victoria, Victoria, BC V8P 5C2

[12]Department of Geology and Geophysics, University of Wyoming, Laramie, WY 82071

[13]Marlboro Institute for Liberal Arts and Interdisciplinary Studies, Emerson College, Boston, MA 02116 and

[14]Department of Earth Sciences, Montana State University, Bozeman, MT 59717

*Correspondence email: adawson@mtroyal.ca

†These authors contributed equally to the development of this manuscript



## 30  Abstract

Land cover governs the biogeophysical and biogeochemical feedbacks between the land surface and

atmosphere. Holocene vegetation-atmosphere interactions are of particular interest, both to understand the climate effects of intensifying human land use and as a possible explanation for the Holocene
Conundrum, a widely studied mismatch between simulated and reconstructed temperatures. Progress has been limited by a lack of data-constrained, quantified, and consistently produced reconstructions of
Holocene land cover change. As a contribution to the Past Global Changes (PAGES) LandCover6k Working Group, we present a new suite of land cover reconstructions with uncertainty for North America,
based on a network of 1445 sedimentary pollen records and the REVEALS pollen-vegetation model coupled with a Bayesian spatial model. These spatially comprehensive land cover maps are then used to
determine the pattern and magnitude of North American land cover changes at continental to regional scales. Early Holocene afforestation in North America was driven by rising temperatures and
deglaciation, and this afforestation likely amplified early Holocene warming via the albedo effect. A continental-scale mid-Holocene peak in summergreen trees and shrubs (8.5 to 4 ka) is hypothesized to
represent a positive and understudied feedback loop among insolation, temperature, and phenology seasonality. A last-millennium decrease in summergreen trees and shrubs with corresponding increases in
open land likely was driven by a spatially varying combination of intensifying land use and neoglacial cooling. Land cover trends vary within and across regions, due to individualistic taxon-level responses to
environmental change.  Major species-level events, such as the mid-Holocene decline of eastern hemlock, may have altered regional climates. The substantial land-cover changes reconstructed here support the
importance of biogeophysical vegetation feedbacks to Holocene climate dynamics. However, recent model experiments that invoke vegetation feedbacks to explain the Holocene Conundrum may have
overestimated the land cover forcing by replacing Northern Hemisphere grasslands >30°N with forests; an ecosystem state that is not supported by these land cover reconstructions. These Holocene
reconstructions for North America, along with similar LandCover6k products now available for other continents, serve the Earth system modeling community by providing better-constrained land cover
scenarios and benchmarks for model evaluation, ultimately making it possible to better understand the regional- to global-scale processes driving Holocene land cover dynamics.



# 1. Introduction

Vegetation is the great mediator of biogeophysical and biogeochemical interactions between the land
surface and the atmosphere (Bonan and Doney, 2018; Harrison et al., 2020; Pongratz et al., 2010; Gaillard

et al., 2010). Enhanced carbon uptake and sequestration by terrestrial ecosystems is an essential
component to contemporary negative-net $CO_2$ emission scenarios needed to stabilize the climate system

and mitigate the dangerous impacts already emerging (Rogelj et al., 2018; van Vuuren et al., 2017).
During the Holocene, as cryosphere-ocean-atmosphere feedbacks waned and anthropogenic land use

intensified (Ruddiman, 2013; Stephens et al., 2019), vegetation-atmosphere feedbacks and forcings
increased in importance, particularly in regions where climate variability interacted with major changes in

vegetation structure. Examples include soil and vegetation feedbacks that amplified precessional-driven
variations in monsoonal rainfall intensity in North Africa and Asia (Chen et al., 2021; Chandan and

Peltier, 2020), and increases in high-latitude tree cover, which decreased wintertime albedo and increased
temperatures (Williams et al., 2011; TEMPO (Testing Earth System Models with Paleo-Observations),

1996; Foley et al., 1994). Intensified human land use and resulting greenhouse gas emissions may have
delayed Northern Hemisphere late-Holocene cooling and glaciation (Ruddiman, 2003). However, initial

models of global anthropogenic land cover change (ALCC) (Kaplan et al., 2009; Klein Goldewijk et al.,
2011, 2010) over the Holocene were largely unconstrained by paleoecological and archaeological

observations and so differed widely in their estimated size and scope of the anthropogenic footprint. More
recently, Holocene increases in vegetation cover have been invoked to explain the Holocene Temperature

Conundrum, a discrepancy between proxy and model-estimated temperature during the early- to mid-
Holocene (Thompson et al., 2022; Kaufman and Broadman, 2023), but global simulations of vegetation-

climate feedbacks during the Holocene are not constrained by observational data. At subcontinental
scales, data-constrained studies of Holocene climate-vegetation feedbacks in Europe indicate that mid-

Holocene vegetation changes relative to pre-Industrial baselines could have warmed winters in some
areas by 4-6°C in northeastern Europe (Strandberg et al., 2022a, 2014).

Hence, there is an on-going need for comprehensive and accurate proxy-based reconstructions of
past land cover at regional to global extents (Gaillard and Group, 2015; Gaillard et al., 2018, 2010). These

reconstructions can then be used with Earth system models (ESMs) to test hypotheses about the physical,
biological, and anthropogenic processes that drove Holocene climate variability (Harrison et al., 2020).

Fossil pollen records offer the primary observational constraint on past vegetation composition and
structure, with thousands of records now available globally. Efforts to systematically map late-Quaternary

land cover using fossil pollen data and well-defined rulesets began in the late 1990s with the Biome6000
project (Prentice et al., 2000, 2011). Since then, the continental-scale pollen databases launched in the



1980s and 1990s (Grimm et al., 2013) have coalesced along with other paleoecological data into the

Neotoma Paleoecology Database (Neotoma), an international, multi-proxy, curated data resource that

helps tame issues of data heterogeneity through community curation by experts (Williams et al., 2018).

Neotoma thus enables global-scale analyses of past vegetation and climate change (e.g. Mottl et al., 2021;

Herzschuh et al., 2023).

Multiple pollen-vegetation models (PVMs) have been developed to make quantitative inferences

about past vegetation. Some PVMs involve relatively simple but effective transfer functions, such as the

modern analogue technique (Williams et al., 2011) or rule-based systems for classifying land cover

(Prentice et al., 2000; Cruz-Silva et al., 2022; Fyfe et al., 2010).  Others are process-based proxy system

models (Evans et al., 2013) that attempt to represent the processes governing the atmospheric transport

and deposition of pollen, such as the REVEALS and LOVE (Sugita, 2007a, c) or STEPPS (Dawson et al.,

2019a). Efforts continue to test and refine the parameterizations of these models through paired analyses

of pollen assemblages and forest composition at local to landscape scales (Liu et al., 2022).

In response to these scientific needs and opportunities, the Past Global Changes (PAGES)

LandCover6k working group was launched as an international effort (Gaillard and Group, 2015) to

reconstruct vegetation globally for the Holocene. LandCover6k, led by experts typically working at

continental scales, has had the explicit aim of creating vegetation reconstructions that can better constrain

past histories of anthropogenic land use in ESMs and is mostly based on networks of fossil pollen

records. To facilitate the use of these vegetation reconstructions in ESMs, the REVEALS PVM has been

used for all LandCover6k reconstructions, with standard model parameterizations and standard protocols

for pollen data handling. LandCover6k gridded REVEALS reconstructions at the continental scale have

been published so far for Europe (Githumbi et al., 2022a, b; Trondman et al., 2015; Serge et al., 2023) and

China (Li et al., 2023b). However, no comparable REVEALS-based land cover reconstructions are

available for North America, despite a comparable density of fossil pollen records to these other regions

(Stegner and Spanbauer, 2023) and prior regional-scale applications of REVEALS in North America

(Sugita et al., 2010; Chaput and Gajewski, 2018).

REVEALS uses pollen counts, pollen productivity estimates, pollen fall speeds, atmospheric

conditions, and sedimentary basin type and size to estimate vegetation composition for a given time

period. REVEALS accounts for the processes of differential pollen production (determined by pollen

productivity estimates) and dispersal-deposition (determined by the pollen fall speeds, atmospheric

conditions, and sedimentary basin type and size).

While REVEALS reconstructions usually combine information from multiple pollen records, REVEALS

is not explicitly spatial, and so does not support the interpolation of inferences to places with no pollen

records. To address this issue, other researchers have developed a statistical approach to spatially



interpolate REVEALS-based land cover estimates from individual grid cells to all cells within the grid,
including estimates of uncertainty (Pirzamanbein et al., 2014, 2018a). The approach uses a Bayesian

hierarchical model with spatial dependence specified according to a Gaussian Markov Random Field
(GMRF; (Lindgren et al., 2011); we refer to the two-step process of REVEALS-based estimation

followed by this spatial interpolation as the REVEALS-GMRF approach. REVEALS-GMRF has been
used to develop spatially continuous gridded vegetation reconstructions in Europe to assess vegetation-

climate feedbacks resulting from natural and anthropogenic land cover change (Strandberg et al., 2022b)
and to evaluate ALCC models (Kaplan et al., 2017) and dynamic vegetation models (Pirzamanbein et al.,

2020; Dallmeyer et al., 2023; Zapolska et al., 2023).

Here we adopt the REVEALS-GMRF approach to reconstruct land cover changes in North

America during the Holocene. This work relies on 1445 Holocene pollen records drawn from Neotoma
and its constituent database, the North American Pollen Database (NAPD), with a targeted data-

mobilization campaign employed to add more records to the NAPD for western North America. Using
these data, we reconstruct the fractional cover of 32 plant taxa and three land cover types (LCTs):

evergreen trees and shrubs (ETS), summergreen trees and shrubs (STS), and open vegetation/land (OVL,
including grasses, herbs, and low shrubs) from 12,000 years ago (12 ka) to present. We present the

Holocene vegetation reconstructions by working across scales, first describing continental-scale trends in
land cover, then shifting to several regional-scale case studies to show how the continental-scale trends

emerge from taxon-level dynamics that vary within and among regions, with respect to key taxa, drivers,
and resultant land-cover changes. We then zoom out to discuss the continental-scale drivers of Holocene

land cover change in North America and possible biophysical implications of these changes for Holocene
vegetation-atmosphere interactions and the Holocene Conundrum. Lastly, we discuss the potential

limitations of the REVEALS-GMRF approach and the opportunities now available for well-constrained
hemispheric- to global-scale studies of vegetation-atmosphere interactions.

## 2. Data and methods

### 2.1 Pollen data and data mobilization for western North America

Western North America has traditionally been underrepresented in the NAPD and Neotoma, but the
density of fossil pollen records in western North America has steadily increased in recent decades, as

multiple teams have worked to collect new records, often focusing on interactions among past vegetation,
fire, climate, and human dynamics (Anderson et al., 2008; Gavin and Brubaker, 2014; Marlon et al.,

2012; Iglesias et al., 2018; Alt et al., 2018). Many of these datasets were contributed to Neotoma



(Williams et al., 2018) when originally published, while others were contributed to Neotoma for an open-data mobilization campaign conducted for this paper and PAGES LandCover6k (Gaillard et al., 2018).

After this effort, 1582 North American Holocene pollen records were downloaded from Neotoma (Supp. Table 1). Each record included pollen count data at a series of depths. For methodological consistency (Flantua et al., 2023), we refit all age-depth models using a custom-built workflow (https://github.com/andydawson/bulk-bchron that assessed chronological constraints and then used these chronological constraints and IntCal20NH (Reimer et al., 2020) to fit the Bchron age-depth model (Parnell et al., 2008). This resulted in 1445 records with age-depth models. Ages of the youngest and oldest chronological constraints were used to determine the reliable age range for each record; we limited extrapolation of pollen sample ages beyond the youngest or oldest constraints to 1000 years. Pollen types were aggregated to taxa using the taxa list in the North American Modern Pollen Database (Whitmore et al., 2005). We used a subset of 32 taxa for this analysis, choosing the most abundant taxa, several open-land indicators, and taxa with available estimates of relative pollen productivity (Tables 2, 5).

## 2.2 REVEALS (regional reconstructions)

Pollen-based land cover reconstructions were performed using the REVEALS pollen-vegetation model (Sugita, 2007b), and based on the standard protocol for PAGES LandCover6k (Trondman et al., 2015; Githumbi et al., 2022a). This model estimates the relative abundance of plant taxa, along with the standard error of these estimates, given pollen counts and input parameters that represent sedimentary basin size and type, pollen productivity estimates (PPEs), pollen fall speeds, and atmospheric conditions. REVEALS was developed to operate at the regional scale (Sugita, 2007a; Hellman et al., 2008a, b); inferences of plant relative abundance represent the background vegetation over large areas (suggested as 100 km x 100 km in (Hellman et al., 2008b), but this scale is variable). REVEALS traditionally has been used to infer plant relative abundance from records from large lakes (>50 ha), but has been tested and applied to regions with records from a number of smaller lakes. The REVEALS model accounts for both differential productivity and dispersal among taxa. Differential productivity is determined by taxon-specific PPEs, while dispersal is modeled according to a Gaussian plume (Sutton, 1953) or Lagrangian dispersal-deposition model (Kuparinen et al., 2007), both of which require the specification of atmospheric conditions including wind speed. REVEALS accounts for differential dispersibility among taxa using pollen fall speeds. See (Sugita, 2007b; Githumbi et al., 2022a) for a more detailed and theoretical description of REVEALS. We implement REVEALS using the REVEALSinR R package (Theuerkauf et al., 2016). REVEALS estimates for other regions included in the LandCover6k effort (Li et al., 2023a; Githumbi et al., 2022a) were developed using more traditional implementations of this

model (LRA.REVEALS.v6.2.4.exe (Sugita, unpublished) and LRA R package (Abraham et al., 2014)),
        which differ slightly in their calculation of the standard errors of relative abundances.

Pollen source areas and the relative representation of plant taxa in the REVEALS dispersal-
        deposition model are affected by sedimentary basin type (e.g. lake, mire) and area (Sugita, 2007a;

Trondman et al., 2016). Basin type is typically included in Neotoma metadata for pollen datasets, but not
        all datasets include metadata on basin area. To determine basin area for these datasets, we developed a

standard workflow (Goring, 2021); https://github.com/NeotomaDB/neotoma_lakes).  First, we used
        hydrological databases (National Hydrography Dataset (United States Geological Survey, 2022), National

Hydro Network (Natural Resources Canada, 2022)) to assign basin areas to datasets whose coordinates
        fell within a water-body polygon. Second, for dataset coordinates that landed outside a water-body

polygon, we used Google Earth Engine to identify the basin. These basins were traced using the polygon
        tool in Google Earth Engine, and basin area was calculated from polygon area. Not all basins could be

identified, however, particularly for pre-GPS sites in Neotoma with imprecise coordinates. Third, for sites
        still without basin area, we assigned a size of 50 ha. In the context of the REVEALS model, this

represents a medium-sized lake. This decision avoids the potential biases from assigning large or small
        lake areas, although site-level reconstructions may over- or under-represent taxa if basin area (and hence

pollen source area) is inaccurate (Jackson, 1990; Davis, 2000; Liu et al., 2022).  All basin areas recovered
        in the first and second steps were added as site-level metadata to Neotoma, along with dataset notes.

We used PPE and fall speed datasets from Wieczorek and Herzschuh (2020) for the Northern
        Hemisphere extra-tropics. Specifically, we used the North America continental-scale datasets, which

include PPE (with grass as the reference taxon) and fall speed values for 30 of the 32 taxa we consider in
        this work. For *Ambrosia* (ragweed) and *Tsuga* (hemlock), which were not included in Wieczorek and

Herzschuh (2020), we use PPE and fall speed values from a previously compiled North America dataset
        (Dawson et al., 2016; Trachsel et al., 2020). Additionally, for the *Larix* (larch) fall speed, we used the

value for *Larix laricina* from (Niklas, 1984), which is the dominant species in eastern and northern North
        America. This estimate is an order of magnitude smaller than the *Larix* taxon-level fall-speed estimate

included in (Wieczorek and Herzschuh, 2020),which originates from (Bodmer, 1922)). We experimented
        with fall-speed datasets that included these larger *Larix* fall-speed estimates; these vegetation

reconstructions indisputably overrepresented larch.
                We used the Gaussian plume dispersal model with a wind speed of 3 m/s and neutral atmospheric

conditions (vertical diffusion coefficient cz=0.12; turbulence parameter n=0.25; wind speed u=3 m/s; see
        (Jackson and Lyford, 1999a)), to be consistent with the dispersal model specified in the LandCover6k

protocol for Northern Hemisphere reconstructions (Dawson et al., 2018; Githumbi et al., 2022a). We set





the region cutoff to the REVEALSinR function default of 100 km; this specifies the maximum distance
that most pollen will originate from (Sugita, 2007a; Theuerkauf et al., 2016).

We reconstructed land cover for 25 time intervals that cover the Holocene. These time intervals
were defined according to the LandCover6k working group protocol (Trondman et al., 2015) (SI Table
4). Time intervals are specified in kiloyears before present (ka), where present is defined as 1950 CE, but
for time intervals <1ka we also note the Common Era (CE) timescale. Intervals in the period from 11.7 to
0.7 ka have a 500-year temporal grain (11.7 to 11.2 ka; 11.2 to 10.7 ka; etc.), while the three most recent
intervals have a finer temporal grain (0.7-0.35 ka [1250-1600 CE]: 350 years; 0.35-0.1 ka [1600-1850
CE]: 250 years; 0.1-(-0.074) ka: 174 years [1850-2024 CE]) in order to better capture the changes
associated with intensifying anthropogenic land use over the last five centuries. Pollen samples were
assigned to a time interval based on their mean calibrated radiocarbon age.  If multiple samples for a
record fell within the same interval, pollen counts were summed by taxon, so that each record would have
at most one set of pollen counts per time interval. We used a grid resolution of 1°x1°, also as specified by
the LandCover6k protocol. Within a grid cell, REVEALS reconstructions for multiple sites were
averaged, as is standard practice (Sugita, 2007a). Taxon-level reconstructions were aggregated to three
land cover categories (SI Table 5): summergreen trees and shrubs (STS), evergreen trees and shrubs
(ETS), and open vegetation/land (OVL), which includes grasses, herbs, and low-stature shrubs. All trees
and shrubs (ATS) is calculated as the sum of STS and ETS.
**2.3 REVEALS-GMRF (spatial modeling and interpolation with uncertainty)**

Here, we use the REVEALS-GMRF Bayesian hierarchical model (Pirzamanbein et al., 2018b) to spatially
interpolate the REVEALS-based land cover reconstructions. REVEALS-GMRF exploits the spatial
dependence in land cover using a Gaussian Markov Random Field, and permits the characterization of
uncertainty given the empirical land cover product. As in Pirzamanbein et al. (2018), we use elevation as
a covariate. Although including simulations of land cover as an additional covariate can further improve
the reliability of resulting land cover maps (Pirzamanbein et al., 2020), the intent of our work is to
develop a spatial land cover product suitable for validation of and assimilation with dynamic vegetation,
land use, and Earth System models. Hence, to maintain independence, we refrain from using simulated
land cover as a covariate. To quantify overall uncertainty of a grid cell for a specified time period, we
computed the area of confidence regions (CR; (Pirzamanbein et al., 2018a)). Smaller CR values indicate
higher confidence, while larger CR values indicate more uncertainty. As in Pirzamanbein et al. (2018), a
CR threshold is determined using the complete set of CR values. Accordingly, any grid cells with a CR
greater than a threshold of 9 were omitted from spatial reconstructions (Githumbi et al., 2022c).  The



result of this spatial interpolation is a set of empirically-based land cover maps for North America and the
       LandCover6k time intervals, with uncertainty.


**2.4 Calculation of proportional changes and mapped anomalies**

To summarize continental-scale land-cover dynamics through time, we built time series of both mean
       relative cover and area-weighted relative cover. We used recently updated ice sheet maps (Dalton et al.,

2020) to identify glaciated and unglaciated grid cells for each time period, then calculated the fraction of
       unglaciated land cover for each time period.  We then calculated the mean relative cover of each land

cover type for each time period, across all unglaciated grid cells at that time period. Mean relative cover
       usefully summarizes land cover change within ice-free regions, but does not track the overall increases in

vegetated land area in North America during the last deglaciation.  To calculate area-weighted relative
       cover, for each time period, we multiplied the relative cover of each grid cell by the cell area, and then

summed the relative cover of each land cover type across space for unglaciated land grid cells. We then
       divided these summed area-weighted cover values for each time period by the sum of total unglaciated

land area at 0.25 ka.  Because the number of unglaciated land grid cells changes through time, while the
       denominator is constant, this area-weighted metric of relative cover is affected by both available land and

changes in land cover proportion, resulting in a proportional metric that is sensitive to continental-scale
       increases in vegetation cover.

For each pair of time intervals, we calculated land cover change as grid-cell scale differences for
       each of the three land cover classes. After visually identifying areas of large land cover changes (Supp.

Fig. 3), we identified several regions for further investigation:  the northeastern US and southeastern
       Canada (NEUS/SEC), eastern Canada (ECAN), western Canada and Alaska (WCAN/AK), and the

Pacific Coast, Cascades, and Sierra Nevada (PCCS) based on areas of greatest change and pollen-site
       density. We then assessed vegetation change for each region at both the taxon and land cover scales,

using the REVEALS grid-cell estimates of taxon mean abundances and the REVEALS-GMRF
       interpolated estimates of mean cover for the land cover types.

3. Results

**3.1 Data coverage**

The assembled dataset of 1445 fossil pollen records has good coverage across the continental US and
       Canada (Fig. 1a).  Areas of relatively high site density include the Great Lakes region of the US and

Canada, the northeastern US, the Rocky Mountains, the Pacific Coast, and central Alaska.  Given good
       data-mobilization efforts for the US and Canada, this distribution reasonably approximates the true



distribution of fossil pollen records, so spatial gaps usually represent lack of available sites (e.g. few lakes

or wetlands in arid regions) or inaccessibility (e.g. high Arctic). Conversely, a lower site density in

Mexico and Central America is partially due to less extensive open-data mobilization efforts in these

regions. Temporally, the distribution of oldest samples (an indicator of record length) is smooth (Fig.

1b), with rapid accumulation of sites between 11.7 and 11 ka (33% of sites have an oldest sample in this

interval) and between 0.5 and -0.074 ka (74% of sites have a youngest sample in this interval).








Figure 1: The spatiotemporal distribution of fossil pollen data used here. A) Map indicating the spatial distribution of sites recovered from the Neotoma Paleoecology Database, as well as the case study regions (WCAN/AK: Western Canada and Alaska; PCCS: Pacific Coast, Cascades, and Sierra Nevada; ECAN: Eastern Canada; and NEUS/SEC: North-Eastern US and South-Eastern Canada). B) Temporal extent of




all sites, in which each site is represented by a horizontal gray line between the site's oldest and youngest

Holocene samples. Sites are ordered by age of the oldest sample, reported as ka. Samples older than 11.7

ka are not shown.

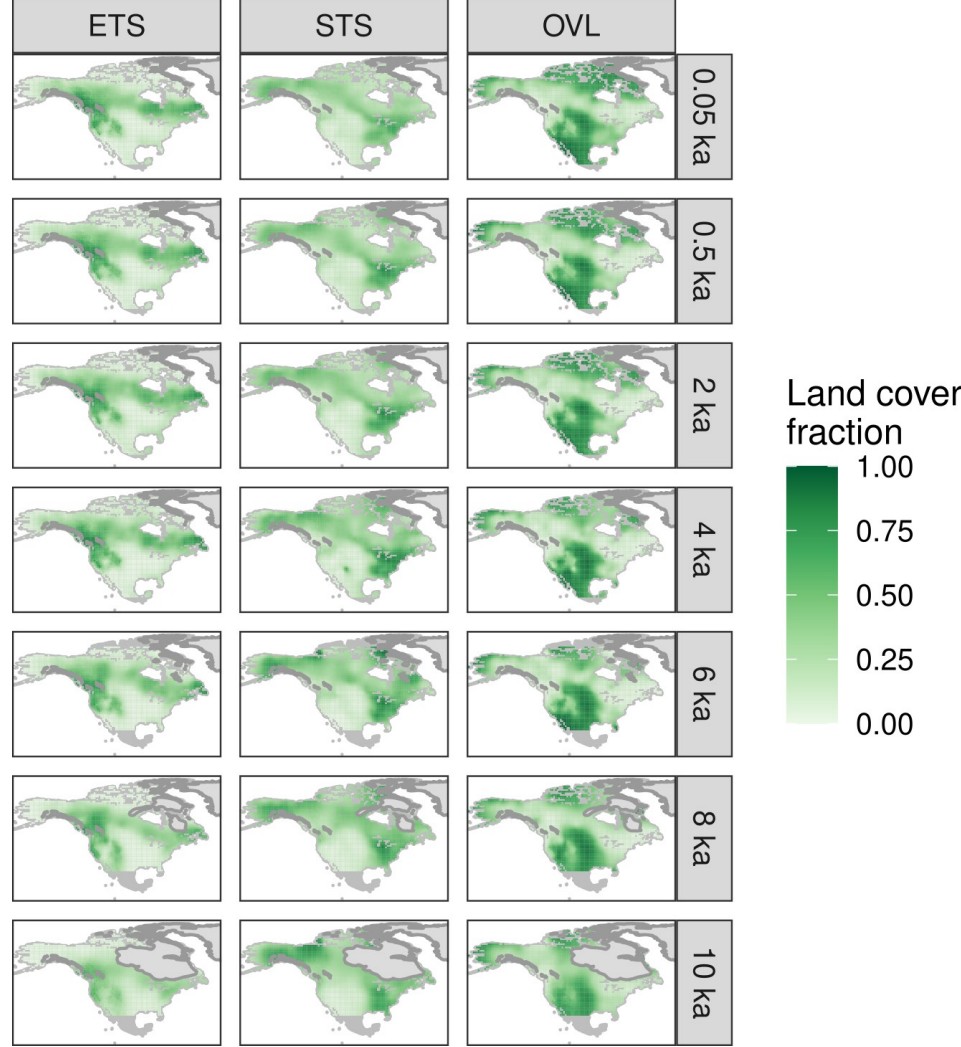


**Figure 2**: Interpolated REVEALS-based estimates of fractional cover for evergreen trees and shrubs

(ETS), summergreen trees and shrubs (STS), and open land (OVL). Laurentide and Cordilleran Ice Sheet

extents (Dalton et al., 2020) are indicated by gray polygons Estimates are presented on a 1°x1° grid, for

selected time periods, with ages reported as ka. Map ordering follows the geological convention of oldest





maps at bottom. The spatial domain for interpolation includes all unglaciated locations in North America
       between 17 and 79°N.


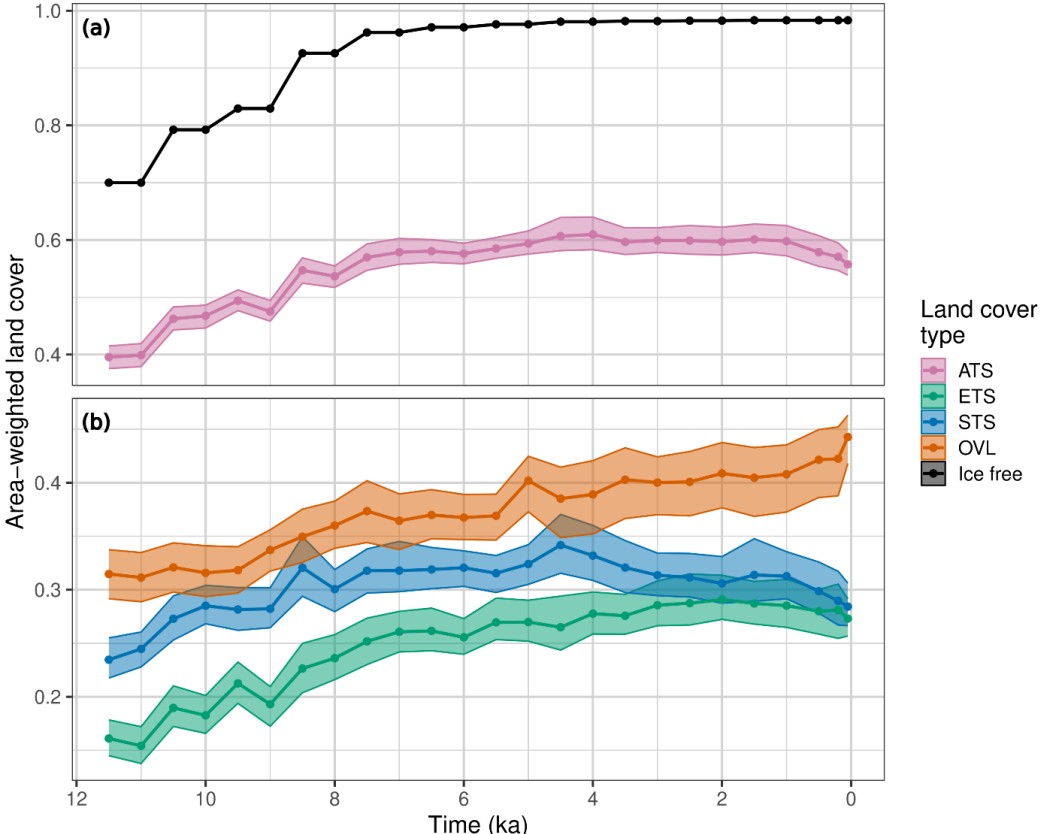


**Figure 3**: (a) Holocene trends in area-weighted cover of all trees and shrubs (ATS) in North America

(pink curve with 95% uncertainty envelope), expressed relative to present unglaciated land area (see
       Methods), and the fraction of unglaciated land relative to continental land area (black). Total cover is the

sum of the evergreen and summergreen trees and shrubs. (b) Trends in the area-weighted cover of
       evergreen (ETS), summergreen (STS) and open vegetation/land (OVL) for North America. For both plots,

land cover estimates are based on the interpolated data for the spatial domain shown in Fig. 2.

**3.2 Continental-scale trends in Holocene land cover**





At a continental scale, the spatial configuration of land cover in North America has been broadly stable during the Holocene (Fig. 2). Persistent features include belts of high evergreen tree cover in the mountainous West and Canada, high summergreen tree cover in the eastern US, moderate to high cover of summergreen trees and shrubs in Alaska and northern Canada, and high proportions of open vegetation in the Great Plains, western Alaska, and Arctic Canada (Fig. 2).

However, despite this broadly stable spatial configuration, there were large continental-scale changes in the area-weighted fractional cover of land cover types in North America during the Holocene, particularly during the early Holocene (Fig. 3a, Supp. Fig. S3). From 11.7 to 7 ka, the area-weighted fraction of forested land cover increased from about 56 to 91%. This increase closely tracked the increase in available land surface area, as the Laurentide Ice Sheet retreated (Fig. 3a) (Dalton et al., 2020). During the early Holocene, the largest gains in forest cover were across deglaciated western Canada and ice-marginal areas in eastern Canada (Fig. 2, Supp. Fig. S3). Forest cover continued to expand between 7.5 and 4.5 ka (Fig. 3a), even though the Laurentide Ice Sheet had disappeared by ~6 ka, with the largest afforestation in the northern Great Plains of central Canada and in recently deglaciated regions of eastern Canada (Fig. 2, Supp. Fig. S3). Continental-scale forest cover remained stable from 4.5 to 1.5 ka, then declined after 1.5 ka (Fig. 3a). The late-Holocene decline in forest cover was most pronounced in the eastern US and in Arctic and boreal Canada where open vegetation began increasing after 4 ka (Fig. 2, Supp. Fig. S3, S4).

Within these Holocene trends, the three land cover types followed differing trajectories (Fig. 3b). All three show a strong increase between 11.7 and 7.5 ka in their area-weighted relative cover, again tracking ice retreat. After 7.5 ka, however, the three trajectories diverged. Evergreen trees and shrubs continued to rise slowly but steadily from 7.5 to 2 ka, then declined slightly (Fig. 3b). Summergreen trees and shrubs reached peak area-weighted cover at 4.5 ka, then declined, with an accelerated decline after 1.5 ka. The proportion of open lands remained largely stable from 7.5 to 4.5 ka, then increased, with an accelerating increase after 2 ka. A close examination of the continental-scale anomaly maps (Supp. Figs. S3, S4) suggests a fairly complex spatial mosaic for each land cover type, with the continental-scale trends emerging from a welter of regional-scale phenomena. For example, widespread gains in evergreen trees and shrubs across much of Canada and Alaska between 6 and 4 ka were partially offset by large losses in the eastern US and southeastern Canada over the same period (Supp Fig. S3). Because different plant species predominate in different regions of North America, these continental-scale trends in land cover were the emergent outcomes of individualistic species-level responses to changing climates and, in some places, intensifying land use (Williams et al., 2004).



### 3.3 Regional case studies

#### *3.3.1 Northeastern US & Southeastern Canada (NEUS/SEC)*

For the mostly forested NEUS/SEC, after initial afforestation and loss of open lands during the early Holocene (10 to 8 ka), the central dynamic has been shifts in the relative cover of evergreen and summergreen trees and shrubs (Fig. 4a, Supp. Fig. S5). The fractional cover of evergreen trees and shrubs declined throughout the early to middle Holocene (10 to 4 ka) in the southern part of the domain, which intensified to widespread loss across the NEUS/SEC (6 to 4 ka), but then reversed after 4 ka, with recovery and regrowth of evergreen trees and shrubs (Fig. 4a). The western increase in open lands between 8 and 4 ka is caused by the eastward expansion of prairie due to drier conditions (Williams et al., 2009b), while the decrease in open lands from 4 to 0.5 ka is due to increased moisture availability that resulted in a westward shift of the prairie-forest border and increase in summergreen forest taxa in the eastern Midwest (Umbanhowar et al., 2006). Overall, the low proportions of open vegetation in the NEUS/SEC from 9 ka until European settlement (Fig. 4B) likely represents these western prairies and local wetlands, which expanded in the late Holocene in parts of the NEUS/SEC (Brugam and Swain, 2000; Ireland and Booth, 2010), rather than grasslands or other open lands in eastern deciduous forests (Faison et al., 2006).

At a taxon level, most changes in evergreen cover can be attributed to regional declines of cold-tolerant conifers such as *Picea glauca*, *P. mariana*, and *P. rubens* (white, black, and red spruce); *Pinus banksiana*, *P. resinosa*, and *P. strobus* (jack, red, and white pine), and *Abies balsamea* (balsam fir) between 10 and 6 ka (Fig. 4c, Supp. Fig. S6) (Spear et al., 1994; Jackson and Whitehead, 1991; Jackson et al., 1997). These changes, combined with the zonal pattern of evergreen expansion in the northern NEUS/SEC and declines in the southern part, suggest that much of the evergreen tree and shrub cover changes during the early to middle Holocene can be attributed to postglacial northward shifts in tree distributions in response to rising temperatures and deglaciation.

A second major feature is the well-known and dramatic expansion, collapse, and re-expansion of *Tsuga canadensis* (eastern hemlock), a cool-temperate conifer that typically occupies warmer climates than *Abies balsamea* (Thompson et al., 1999). Investigations continue into understanding the abrupt and widespread collapse in *Tsuga canadensis*, which differed from the overall evergreen trend. The collapse occurred in less than 10 years at some sites (Allison et al., 1986) and is linked to regional shifts in water availability and temperature gradients (Booth et al., 2012; Foster et al., 2006; Haas and McAndrews, 1999; Shuman et al., 2023). Initial hypotheses that a pest or pathogen such as hemlock looper (Bhiry and Filion, 1996; Davis, 1981; Anderson et al., 1986) caused the hemlock decline have not been supported by recent investigations (Oswald et al., 2017), although insect remains are scarce in lacustrine archives. Understanding the causes of the hemlock collapse is outside the scope of this paper; however, this paper





shows its importance, in that a single-species collapse fundamentally altered the functional composition, ecosystem phenology, and land cover of the NEUS/SEC for thousands of years.
Among summergreen taxa, these REVEALS reconstructions indicate a strong growth in *Acer* (maple) cover in the NEUS/SEC until 4 ka and declines thereafter. *Acer saccharum* (sugar maple) is
probably the dominant taxon driving this curve, with *A. rubrum* (red maple) increasing in the late Holocene (Finkelstein et al., 2006). *Quercus* (oak) abundances were high between 9 and 4.5 ka, then
steadily declined, while *Betula* (birch) abundances remained relatively stable. *Fagus grandifolia* (American beech) abundances continued to steadily increase throughout this period until 3 ka, then began
declining after 1.5 ka, along with *Tsuga canadensis*, while *Abies balsamea* abundances increased. Disturbance-related taxa and indicators of open vegetation such as *Ambrosia* (ragweed) and *Rumex*
(sorrel*, not shown) begin increasing at ca. 0.35 ka (1600 CE). These late-Holocene changes in forest composition can be plausibly attributed to regional cooling (explaining the increase in *Abies balsamea*)
and perhaps also intensified human land use and disturbance, with the relative importance of these drivers varying at subregional scales and among taxa (Oswald et al., 2020; Mottl et al., 2021).
An intriguing feature of these REVEALS reconstructions is the inference of *Abies balsamea* and *Acer* spp. as the most common evergreen and summergreen tree taxa in the NEUS/SEC, given that *Picea*,
*Pinus*, and *Quercus* are more abundant in pollen and macrofossil records and prior site-level and regional syntheses of Holocene pollen records have emphasized their dynamics (Jackson et al., 1997; Jackson and
Whitehead, 1991; Payette et al., 2022; Spear et al., 1994). Witness-tree data in the NEUS also indicate that *Fagus* and *Quercus* were the most abundant broadleaved taxa at time of European settlement
(Thompson et al., 2013). One possible reason for the higher levels of *Abies* and *Acer* reported here is that our NEUS/SEC domain extends a bit farther north than prior reconstructions that have focused more
on central and southern New England. A second possibility is that the parameterizations for *Abies* and *Acer* are incorrect, causing the coverages of these taxa to be overestimated (see Discussion).





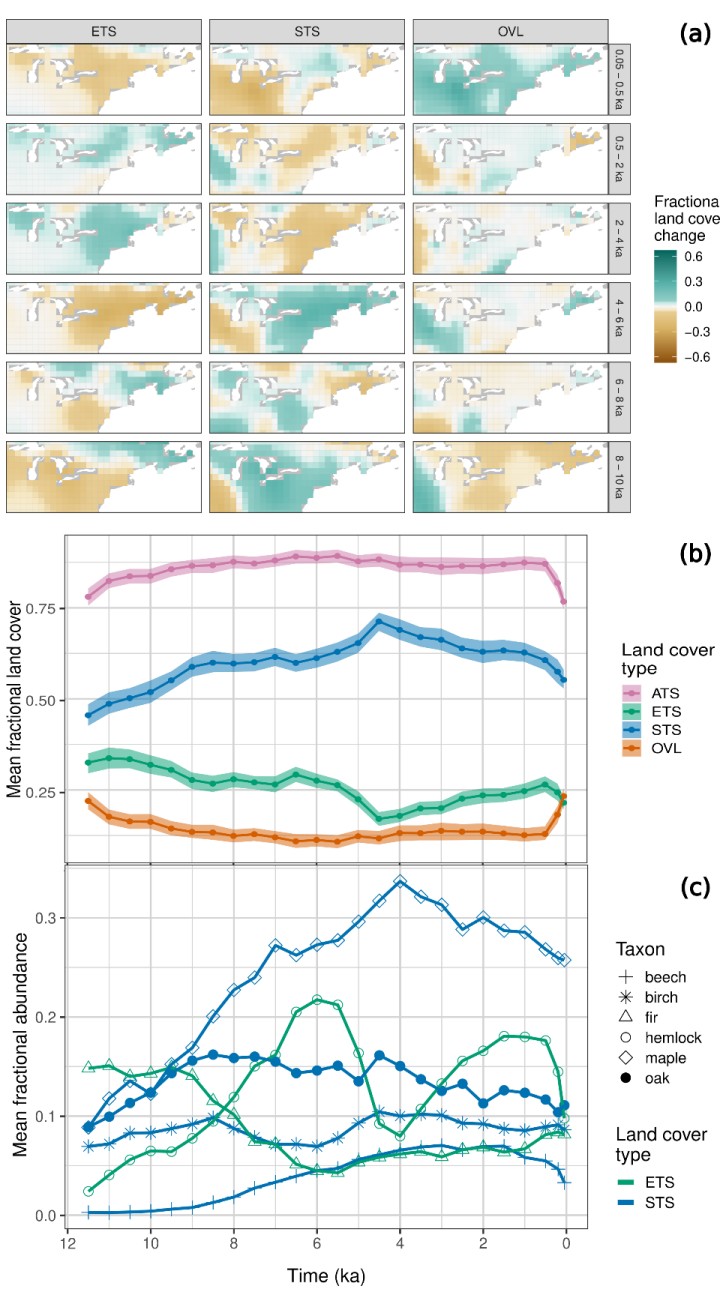

**Figure 4:** (a) Land cover anomaly maps for the northeastern US and southeastern Canada (NEUS/SEC) case-study region. Maps show the anomalies in fractional cover for each land cover class for pairs of indicated time intervals. Spatial resolution is 1°x1° and time units are ka. (b) Holocene trends in the mean relative cover of the three land cover types (ETS, STS, OVL)





and the ATS sum for the uninterpolated REVEALS grid cell estimates across the NEUS/SEC
region. (c) Holocene trends for the REVEALS abundance estimates for the six most commonly
occurring taxa in the region.  Line color indicates assignment of individual taxa to land cover
types. For taxon-level maps, see Supp. Fig. 6.

### *3.3.2 Eastern Canada (ECAN)*

In Eastern Canada (ECAN), immediately north of the NEUS/SEC, evergreen trees and shrubs expanded
across the entire region until 4 ka, with slower and more spatially heterogeneous expansion of evergreen
cover between 4 and 2 ka (Fig. 5a, Supp. Fig. S7).  After 4 ka, open vegetation expanded (Fig. 5a, 5b).
This expansion of evergreen trees and shrubs can be traced to separate phases of expansion for *Abies
balsamea* and *Picea*, with *A. balsamea* expansion mostly between 11.5 and 7.5 ka (in the southern
portion) and *Picea* expansion mostly between 7.5 and 3 ka, across the area and especially in the central
portion (Fig. 5c, Supp. Fig. S8).  Increasing Cyperaceae abundance after 3.5 ka signal the expansion of
forest-tundra.  Major hardwood summergreen taxa include *Betula* and *Alnus*.  *Alnus* abundances expanded
until 6 ka, then slowly declined, while birch abundances expanded between 8.5 and 6 ka, then also
declined.  These declines in *Alnus* and *Betula* after 6 ka co-occurred with expansions of *Picea* and
Cyperaceae, suggesting that the regional vegetation shifted from more of a mixed forest or woodland in
the early Holocene, to evergreen coniferous forest, forest-tundra, or tundra  in the middle to late
Holocene. Note that *Populus* is not included in these REVEALS reconstructions, but it is an important
taxon in ECAN (Peros et al., 2008), and its omission may cause an underestimate of summergreen cover.

 These vegetation changes in ECAN can be attributed to a combination of deglaciation, changing
temperatures, and fire history.  The Laurentide Ice Sheet collapsed at 8.4 ka and the last remnants of the
Labrador Dome disappeared from northern Quebec by 5.7 ka (Dalton et al., 2020).  The replacement of
open land by forest cover during the early Holocene was driven primarily by increases in evergreen trees
and shrubs. The open forests at this time lack modern analogues and varied spatially in taxonomic
composition, with more fir and birch to the east and spruce toward the west (Richard et al., 2020). By 7.5
ka, closed forests developed in response to warmer temperatures. This was followed by a decrease in
summergreen taxa in the southern portion of the boreal forest as well as a decline in fir in the lichen
woodland and feathermoss forest toward the west (Fréchette et al., 2021; Richard et al., 2020).  In the
lichen woodlands of the north, shrub birch and paper birch decreased after 6ka, although remaining high
in the southwest (Fréchette et al., 2018). Near treeline, spruce forests were more open between 7 and 4 ka,
with *Alnus* and shrub *Betula* more abundant (Fréchette et al., 2018; Gajewski, 2019). In ECAN, spruce
abundances reached a maximum between 4 and 2 ka (Fréchette et al., 2018), with the southern portion of
the forest tundra becoming lichen woodland at this time.  After 2 ka, *Picea* abundances at northern sites



declined and open lands increased in response to cooling, but the timing and rate of *Picea* losses varied

among sites and is governed in part by fire history (Gajewski et al., 2021; Gajewski, 2019).









**Figure 5:** (a) Land cover difference maps for ETS, STS, and OVL for the eastern Canada (ECAN) case study. (b) Holocene trends in the mean relative cover of the three land cover types (ETS, STS, OVL) and the ATS sum for the uninterpolated REVEALS grid cell estimates across the ECAN region. (c) Holocene trends for the REVEALS abundance estimates for the six most commonly occurring taxa in the region. For both panels, figure conventions follow Figure 4. For taxon-level maps, see Supp. Fig. 8.

### 3.3.3 Western Canada and Alaska (WCAN/AK)

In western Canada and Alaska (WCAN/AK), evergreen trees and shrubs rapidly expanded between 11 and 8 ka, while the coverage of summergreen trees and shrubs and open lands decreased (Fig. 6a, 6b, Supp. Fig. S9). Most of Alaska was not glaciated during the Last Glacial Maximum; however, much of western Canada, the Brooks Range in Alaska, and south-coastal Alaska were covered by ice (Dalton et al., 2020). Evergreen tree species such as *Picea glauca* (white spruce) persisted in this region in local microrefugia (Anderson et al., 2006), then expanded their ranges during end-Pleistocene warming and deglaciation. Other evergreen taxa such as *Pinus contorta* expanded northward with deglaciation (MacDonald and Cwynar, 1991). Trends in taxon abundances differ substantially between the western and eastern subregions (Fig. 6c, Supp. Fig. S10). In the eastern subregion, changes in taxon abundances were muted, with a modest expansion in *Picea* and *Tsuga* between 11.5 and 7.5 ka, and a slow decline in *Betula* between 10 and 6 ka. *Tsuga* (mostly *Tsuga heterophylla*; western hemlock) occurs primarily along the coast, was abundant along the south coastal areas by 11 ka (Lacourse et al., 2012; Lacourse and Adeleye, 2022), arrived by 9.5 ka into southeast Alaska (Hansen and Engstrom, 1996; George et al., 2023), increased in abundance along the south-coastal areas by 8 ka, and then expanded during the mid- to late Holocene into south-central Alaska (Anderson et al., 2017) and the inland mesic forests of British Columbia (Rosenberg et al., 2003; Gavin et al., 2021). In the western part of this domain (mainly Alaska), changes in summergreen hardwood (mostly shrub) taxa predominate, with a major expansion of *Alnus* between 11.5 and 6.5 ka (Anderson and Brubaker, 1994; Cwynar and Spear, 1995), and a modest expansion of *Betula* between 11.5 and 9 ka. These expansions were accompanied by declines in Poaceae and Cyperaceae, with continued decline in Cyperaceae until 6 ka. *Picea* abundances steadily increased until 4 ka. The net effect was a decline in open vegetation and expansion of evergreen and summergreen trees and shrubs during the early to middle Holocene, with apparent region-wide stability after 4 ka (Anderson et al., 2019).

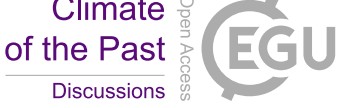

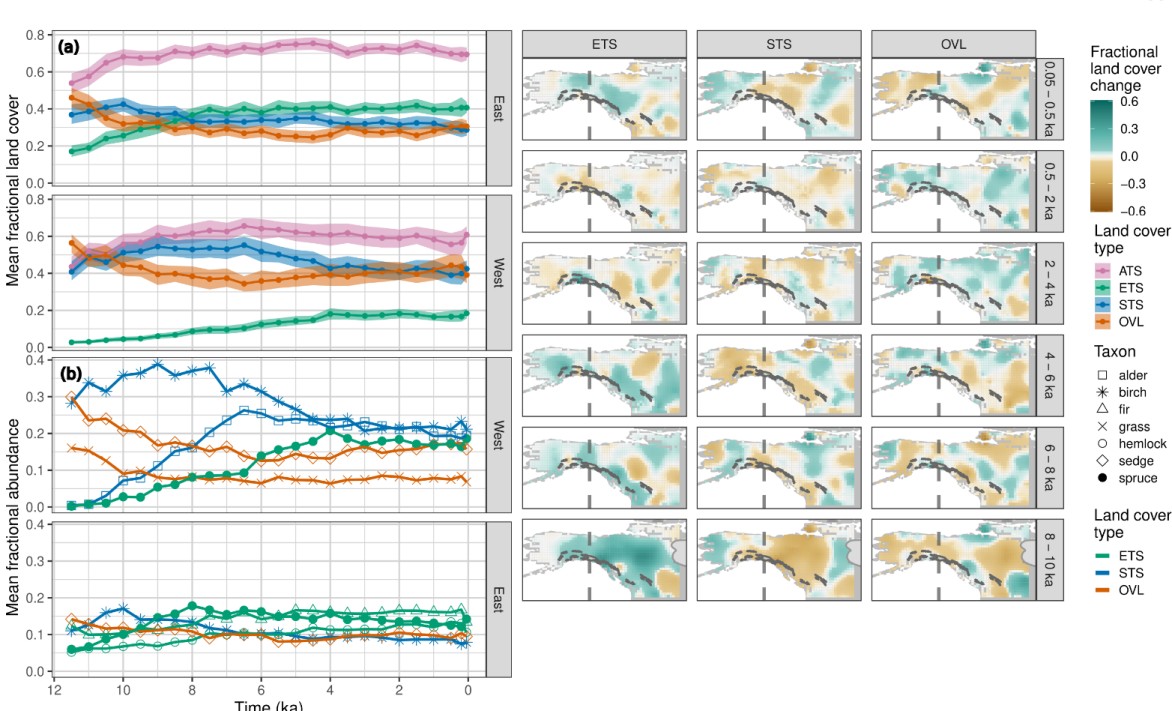

**Figure 6:** (a) Land cover difference maps for the western Canada / Alaska (WCAN/AK) case study region. (b) Holocene trends in the mean relative cover of the three land cover types (ETS, STS, OVL) and the ATS sum for the uninterpolated REVEALS grid cell estimates across the WCAN/AK region, with grid cells averaged separately for eastern and western subregions (dividing line shown in (a) as vertical dashed line). (c) Holocene trends for the REVEALS abundance estimates for the six most commonly occurring taxa in the region, with grid cells averaged separately for eastern and western subregions. For both panels, figure conventions follow Figure 4. For taxon-level maps, see Supp. Fig. 10.

### 3.3.4 Pacific Coast, Cascade, and Sierra Nevada Ranges (PCCS)

In contrast to the other regions, evergreen trees have dominated much of the land cover in the PCCS region since 12 ka (Fig. 7a, 7b, Supp. Fig. S11). In the northern subregion (often referred to as the Pacific Northwest), the proportions of cover types were fairly constant until 9 ka, after which summergreen and open land declined until 6 ka (Fig 7a, 7b). However, taxon-level changes were very dynamic. From 11.7 to 10.5 ka, the REVEALS-estimated abundance of *Pseudotsuga menziesii* (Douglas-fir) more than doubled, reflecting its arrival, expansion, and northward migration (Gugger and Sugita, 2010), and replacing true firs (*Abies*) and *Picea* (Fig 7c, Supp. Fig. S12). In the coastal ranges, frequent forest fires in the early Holocene contributed to a *Pseudotsuga-Alnus rubra* association in the mountains (Gavin et





al., 2013; Long et al., 1998, 2007) and an oak savanna (not shown) that was common in the lowlands (Walsh et al., 2008, 2010; Giuliano and Lacourse, 2023). Summergreen taxa (mostly *Alnus rubra*) and open-land pollen indicators declined from 8 to 6 ka (Fig 7b), replaced by shade-tolerant *Abies*, *Tsuga*, and Cupressaceae, which is mostly western redcedar in this region (Fig. 7b) (Lacourse, 2009; Gavin et al., 2013; Worona and Whitlock, 1995).  After 6 ka, the overall abundance of conifer taxa remained constant (Fig 7a, 7b), although the proportion of hemlock and cedar increased steadily through the late Holocene while *Pseudotsuga* declined, consistent with less fire and the persistence of old-growth forest (Whitlock, 1992; Lacourse and Adeleye, 2022), until the logging of the last century, which manifested as a decline in conifer and increase in summergreen and open land in the coastal forests (Fig. 7a) (Davis, 1973; Whitlock et al., 2018).

The most common taxon in these REVEALS reconstructions, *Abies*, varied little over the last 9 ka, but this genus represents several shade-tolerant species (*A. lasiocarpa* [subalpine fir], *A. grandis* [grand fir], *A. amabilis* [Pacific silver fir], *A. procera* [noble fir]) that collectively are common throughout the region. As noted for the NEUS/SEC, the reconstructed values of *Abies* are likely too high, because REVEALS estimate of fall speed is overestimated.  *Pinus* also represents several species and was most abundant in the dry eastern areas where it was consistently abundant during the Holocene (Minckley et al., 2007).  *Pseudotsuga* and *Larix* have indistinguishable pollen morphologies and are therefore grouped in the reconstructions; however, *Larix* is limited to the eastern edge of this region.

The southern subregion (the Sierra Nevada, Klamath Mountains and California Coast ranges and interior Great Basin) supported roughly equal amounts of conifer forest and open land that overall had minor relative changes over the Holocene (Fig. 7a). Summergreen trees and shrubs have been infrequent contributors to land cover in this subregion. At the taxon-level, *Abies* is again the most common component of reconstructed evergreen land cover across the region, although its abundance may be overrepresented in this parameterization of REVEALS (Fig. 7b). *Pinus* and *Quercus* were  most abundant in the early Holocene, forming open forests and woodlands, especially in the Klamath Mountains and Sierra Nevada (Anderson, 1996; Briles et al., 2008) fire-prone forests east of the Cascade Range (Walsh et al. 2015);.  Most of the eastern portion of this subregion is the Great Basin shrub steppe, where the few pollen records that exist are from high-elevation sites and low-elevation wetlands that are sensitive to fluctuating water tables, recorded as large fluctuations in Cyperaceae pollen. Increases in open-land cover taxa (Cyperaceae, Poaceae, Asteraceae) between 10-7.5 ka and after 3 ka in this region may reflect either local expansion of alpine meadows and wetlands or expansion of steppe more broadly (Mensing et al., 2008; Brugger and Rhode, 2020; Minckley et al., 2007; Thompson, 1992).  Note, however, that desert, steppe, and other open-land arid ecosystems are likely to be underrepresented in these reconstructions, due to a scarcity of dryland sites.




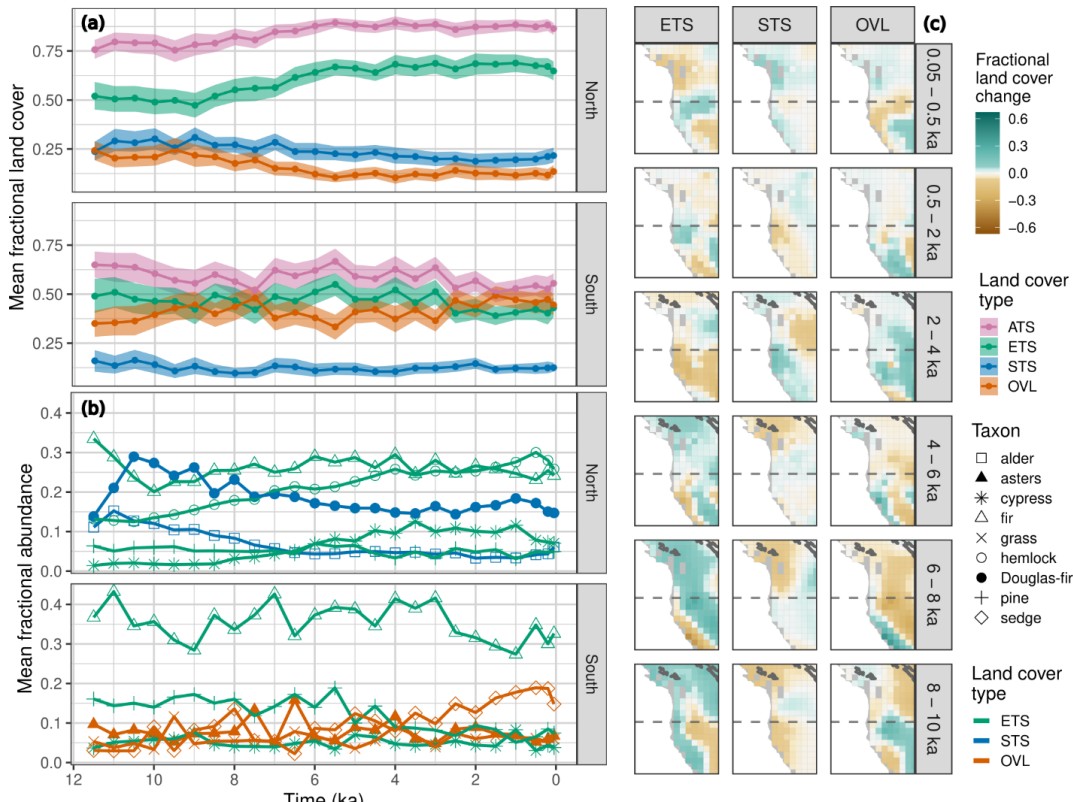

**Figure 7**: (a) Land cover difference maps for the Pacific Coast, Cascade and Sierra Nevada (PCCS) case
study region. (b) Holocene trends in the mean relative cover of the three land cover types (ETS, STS,
OVL) and the ATS sum for the uninterpolated REVEALS grid cell estimates across the PCCS region,
with grid cells averaged separately for northern and southern subregions (dividing line shown in (a) as
vertical dashed line). (c) Holocene trends for the REVEALS abundance estimates for the six most
commonly occurring taxa in the region, with grid cells averaged separately for northern and southern
subregions.  For both panels, figure conventions follow Figure 4. For taxon-level maps, see Supp. Fig.
12.


## 4. Discussion

**4.1 Drivers of Holocene land cover change in North America: scaling from taxon-level regional
dynamics to continental-scale trends**



These continental-scale Holocene changes in land cover (Figs. 2-3) are an emergent outcome of the
individualistic plant responses to deglaciation and multiple environmental changes, including seasonal
temperatures, effective moisture, atmospheric carbon dioxide, soil development, fire and other
disturbance regimes, and, at some locations during the late Holocene, human land use.  As shown here
(Figs. 4-7), this interplay differed spatially across North America, often with opposing trends (e.g.
simultaneous increases in evergreen cover in some regions and decreases elsewhere) that partially offset
at the continental scale.

The first-order continental-scale drivers of Holocene land cover change in North America were
changing climates and the retreat and disappearance of the Laurentide Ice Sheet, particularly during the
early Holocene (11.7 to 7.5 ka) and lasting until 5.7 ka, with the disappearance of remnant ice in the
Labrador Dome in northern Quebec (Dalton et al., 2020). Deglaciation and an overall increase in land
availability explains why all land cover types show an increase in area-weighted cover during the early
Holocene, with evergreen taxa showing the greatest gain (Fig. 3). Continental temperatures also closely
tracked the decline in ice area (Marsicek et al., 2018), and vegetation responded to both factors.
Superimposed upon the early-Holocene increase in area-weighted cover for all PFTs are gains and losses
for each PFT at subregional scales (Figs. 4-7), which can be linked to climate-driven changes in plant
abundances at local to landscape scales that scaled upwards to continental-scale shifts in plant
distributions, with both within-range shifts in dominance and leading-edge and trailing-edge range
dynamics for individual plant taxa (Payette et al., 2022; Williams et al., 2004; George et al., 2023;
Dallmeyer et al., 2022; Anderson et al., 2017).

Leading-edge dynamics and range expansion of tree taxa appear to have been primarily
controlled by the increasing availability of deglaciated land area, end-Pleistocene warming, the declining
influence of ice sheet on regional climates and moisture availability (Alder and Hostetler, 2015; Bartlein
et al., 2014), and on-going expansion of plant taxa into areas of increased climate suitability (Payette et
al., 2022; Williams et al., 2004; George et al., 2023; Dallmeyer et al., 2022).  Declining ice extent favored
moisture advection into eastern areas where most of the increase in summergreen tree taxa took place and
reduced it in midcontinental North America, where most of the open vegetation increase developed
(Shuman and Marsicek, 2016; Shuman et al., 2002; Liefert and Shuman, 2020).  These changes in the
patterns of moisture availability simultaneously favored the large increase from 10 to 6 ka in
summergreen trees and shrubs in eastern North America south of the former Laurentide Ice Sheet and the
increase in open land in the mid-continent (Supplementary Fig. 2-3).

Declines in abundance and trailing-edge dynamics of tree taxa were likely governed by a
combination of declining climate suitability and, in some places, fire regimes, in which many local
populations failed to re-establish after one or more fire events.  For example, in the high northern latitudes



of Quebec and Labrador, the expansion of open lands over the last three thousand years can be attributed first to declining summer insolation and temperatures, caused by precessional changes in the Earth's orbit
(Payette, 2021), but at local to landscape and centennial scales, loss of forest cover was asynchronous and caused by individual fire events (Payette et al., 2008; Gajewski et al., 2021).   Similarly, in the southern
Great Lakes region and NEUS, the late-Pleistocene to early Holocene transition from conifer-dominated forests and parklands to summergreen forests was driven by rising temperatures and changes in effective
moisture (Shuman et al., 2002; 2019), but accelerated locally by intensified fire regimes, with the timing and rate of conversion varying among sites (Jensen et al., 2021; Clark et al., 1996). The expansion and
then retreat of the Great Plains prairies (Fig. 4) was driven by early Holocene aridification and middle- to late-Holocene increases in moisture availability, while fire may have facilitated prairie expansion, but
delayed its retreat (Williams et al., 2009; Nelson et al., 2006; Umbanhower et al., 2006). Of course, throughout North America, forests and fire dynamically co-existed and interacted during the Holocene
(Iglesias and Whitlock, 2020; Kelly et al., 2013), so whether fire causes transformative shifts in ecosystem type depends on its synergistic interactions with directional changes in climate and other
factors (Napier and Chipman, 2021).  Regardless of the role of fire or other disturbances, regional shifts in temperature and moisture availability are usually the first-order predictors of Holocene changes in
vegetation composition (Dean et al., 1984; MacDonald, 1989; Calder and Shuman, 2017; Shuman et al., 2004; Nelson et al., 2006).

The relative proportion of evergreen and summergreen cover types may have been affected by changes in the length and intensity of the growing season during the Holocene, which is regulated by

precessional variations in insolation.  In particular, the broad peak in summergreen tree cover between 8.5 and 3.5 ka (Fig. 3) is consistent with the hypothesis that summergreen tree and shrub abundances in North
America are partially regulated by summer insolation and temperatures (Delcourt and Delcourt, 1994; Williams et al., 2001).  Summer insolation reached higher peak intensity but has a shorter seasonal
duration in the early Holocene than during the late Holocene (Huybers, 2006; Jackson et al., 2009), which may have favored plants with summer deciduous phenology that could more effectively exploit available
energy during a briefer but more intense growing season (Delcourt and Delcourt, 1994; Williams and Jackson, 2007; Edwards et al., 2005). Indeed, pollen-reconstructed and CCSM3-simulated changes in
growing-degree days also peaked during the mid-Holocene when forest cover was greatest, and several millennia after peak summer temperatures, likely because of differential responses of maximum summer
temperatures and total growing season warmth to orbital and other forcings (Marsicek et al., 2018).  This may in turn suggest a seasonal-scale feedback loop between summer insolation, deciduous phenology,
and total summer warmth (see Section 4.2).



During the late Holocene, human land use in the Americas intensified, with increasing effects
upon land cover and understanding the interactions among past climate change, fire regime, and human
land use is a highly active area of research.  Rates of vegetation change worldwide began to increase
between 4.6 and 2.9 ka (Mottl et al., 2021), consistent with growing intensification and extent of land use
(Stephens et al., 2019). The 7 to 10 ppm decrease in global $CO_2$ between 1570 and 1620 CE has been
attributed to the land abandonment and reforestation due to mass mortality of Indigenous populations in
the Americas, caused by the spread of multiple pathogens (Lewis and Maslin, 2015). The best evidence
for dense human populations and extensive land clearance in the Americas comes from Central and
tropical South America (Islebe et al., 1996; McMichael, 2020).

In North America, the effects of human land use prior to EuroAmerican settlement are detectable
at some sites, but are not easily detected at the regional to continental scales addressed here.  In North
America, the distribution of radiocarbon dates from archaeological contexts suggest that population levels
remained relatively low and increasing slowly until ~7 ka, with further increases between 7 and 2 ka, and
then rapidly increased after 2 ka (Peros et al., 2010). Indigenous cultures in North America clearly
engaged in activities that, in some areas, modified the composition and structures of forested and
unforested landscapes for millennia (Ellis, 2021; Delcourt et al., 1986; Leopold and Boyd, 1999; Munoz
et al., 2014). In densely populated regions, such as lands of the Iroquois Confederacy, southern Ontario,
or the Cherokee Nation, Cahokia and the American Riverbottom, land use altered vegetation structure and
composition at local to landscape scales, through management of fire regimes (Roos et al., 2018;
Anderson and Carpenter, 1991), land clearance for agricultural crops (McAndrews and Turton, 2007), and
silviculture, e.g. favoring the spread of nut-bearing trees (Munoz et al., 2014; Black et al., 2006). In
western North America, the effects of anthropogenic activity on vegetation and fire is an active area of
research, with several paleoecological studies indicating changes consistent with human influences on
forest composition (Walsh et al., 2015; Knight et al., 2022; Lacourse et al., 2007).  Because the scale of
human action was strongest at local to landscape scales and varied in intensity within and among regions,
its detection often requires highly-focused, local- to regional-scale studies (Oswald et al., 2020; Roos,
2020; Lacourse et al., 2007; Knight et al., 2022; Munoz and Gajewski, 2010).  These studies clearly
indicate a high intersite variance in the level and detectability of human impact.  At the continental scale,
Gajewski et al. (2019) did not find clear correlations between human population abundance and pollen
abundance, including taxa of economic use or disturbance taxa, and concluded impacts were at local to
regional scales.  Thus, at the regional to continental scales considered here, it is an open question whether
the rich relationships and diverse activities engaged by communities within their homelands led to
detectable changes in land cover types.  We expect that pollen-based land cover reconstructions will



continue to contribute to interdisciplinary approaches to address anthropogenic fire and land cover change during the Holocene (Snitker et al., 2022).

In contrast, after EuroAmerican arrival and expansion from 1492 onwards, North American ecosystems were massively transformed (Stegner and Spanbauer, 2023) by multiple anthropogenic

processes. These include widespread forest conversion to agricultural and pastoral lands, intensive forest harvesting, massive hydrological change through dam construction and wetland drainage, introduction of
exotic species and pests, and both greatly increased fire activity and fire suppression (Klein Goldewijk et al., 2011; Foster and Aber, 2004).

**4.2 Biophysical vegetation-atmosphere feedbacks to Holocene climates**

*4.1.1 Holocene vegetation-atmosphere feedbacks: prior work and need for data constraints*

This paper and the other REVEALS-based reconstructions for the Northern Hemisphere (Githumbi et al.,

2022c; Li et al., 2020) are now laying the foundation for a next generation of Holocene vegetation-atmosphere research with well-constrained land surface data. Many studies have explored the potential
impacts of vegetation-atmosphere feedbacks on Holocene climate dynamics at hemispheric to global scales, but these studies have generally not employed well-constrained land cover reconstructions.
Treeline shifts have been recognized as an important regulator of Holocene vegetation-atmosphere feedbacks in the high northern latitudes (TEMPO (Testing Earth System Models with Paleo-
Observations), 1996). Earth system model experiments with prescribed vegetation scenarios have shown that, in the northern latitudes (45-90N), changes in forest cover are the largest contributor to changes in
net land surface radiation through snow masking and effects on surface albedo (Bathiany et al., 2010). In mid-Holocene atmosphere-vegetation model simulations, strong snow masking resulted in warming three
times higher than those with weak snow masking (Otto et al., 2011). Additionally, empirical estimates of surface-albedo feedbacks in high latitudes are stronger than predicted by climate models (Hogg, 2022).
More recent work has highlighted the importance of the type and density of forest cover in determining the albedo feedback and magnitude of snow masking (Loranty et al., 2014; Alessandri et al., 2021).
Hence, because snow masking and surface albedo is an important regulator of surface-atmosphere feedbacks in climate models, it is also an important source of uncertainty in Holocene climate
simulations, because of the limited availability of well-constrained reconstructions of past changes in vegetation type, structure, and density. Vegetation structure and topographic complexity also jointly
govern surface roughness, which affects lower atmosphere temperature, humidity, wind speed, and soil moisture (Bonan, 2015) and is the dominant land-cover influence on micrometeorological processes
(Chen and Dirmeyer, 2016). This new generation of LandCover6k vegetation reconstructions thus





promise to sharpen our understanding of the role played by surface-atmosphere feedbacks in Holocene
      climate dynamics, with first results from Europe already underway (Strandberg et al., 2022b).

*4.1.2 Assessing recent modeling studies of the Holocene conundrum*

              The Holocene Conundrum has been a major focus of paleoclimatic research over the last decade
(Kaufman and Broadman, 2023; Liu et al., 2014), and recent research has invoked vegetation-atmosphere
      feedbacks to resolve this conundrum (Thompson et al., 2022). The Holocene Conundrum involves a
discrepancy between proxy-based reconstructions of global temperature changes, mostly based on marine
      records, which indicate a mid-Holocene maximum and mid- to late-Holocene cooling (Marcott et al.,
2013), while transient model simulations show small but steady warming throughout the Holocene (Liu et
      al., 2014). Many explanations for this discrepancy have since been proposed (Kaufman and Broadman,
2023). Recent prescribed-vegetation experiments in climate models produce an early Holocene warming
      to a Holocene Thermal Maximum, followed by cooling towards the pre-industrial Holocene (Thompson
et al., 2022), consistent with paleoclimatic proxies. Conversely, experiments that include drivers such as
      dust, ice cover, orbital forcing, and greenhouse gasses without accounting for Northern Hemisphere
vegetation changes do not result in a mid-Holocene thermal maximum (Thompson et al., 2022).

              Our reconstructions suggest, however, that for North America, the prescribed vegetation maps
used by Thompson et al (2022) overstate the magnitude of Holocene vegetation change. These
      prescribed-vegetation experiments for 9 and 6 ka fully replace all $C_3$ grasses with boreal forest for all
locations north of 50N (Thompson et al., 2022, Supp. Fig. 7B). This pattern is qualitatively consistent
      with the early Holocene afforestation reported here (Fig. 2) but is inconsistent with the demonstrated
persistence of tundra throughout the early and middle Holocene, particularly in WCAN/AK (Figs. 2, 6)
      and Canadian High Arctic (Fig. 2). Similarly, the prescribed full replacement of C3 grasslands with
temperate deciduous forest for the 9 and 6 ka experiments (Thompson et al., 2022, Supp. Fig. 7B) is
      inconsistent with clear palynological evidence of the establishment of the Great Plain grassland by the
early Holocene and prairie expansion during the early to middle Holocene (Fig. 2, Supp. Fig. 1) (Williams
      et al., 2009a). Hence, the Thompson et al (2022) simulations should be viewed as useful experiments that
show the potential sensitivity of Holocene climates to large vegetation changes, but these experiments
      likely overestimate the contribution of vegetation feedbacks to resolving the Holocene Conundrum.

      *4.1.3 Understudied Holocene vegetation-atmosphere feedbacks in North America*

Our reconstructions also highlight several major features of North American vegetation dynamics
      that may have underappreciated effects on Holocene vegetation-atmosphere feedbacks and climate
dynamics at regional to continental scales. First, the mid-Holocene decline and recovery of *Tsuga*

*canadensis* (eastern hemlock) in the northeastern United States is an example of how single-species

dynamics can drive fundamental changes in vegetational structure. Although the patterns and drivers of

the *T. canadensis* decline have been extensively studied (Oswald and Foster, 2012; Oswald et al., 2017;

Booth et al., 2012; Foster et al., 2006), the effects of the *T. canadensis* decline on Holocene climates are

unknown. As the dominant evergreen conifer of cool-temperate eastern North America, the decline of *T.*

*canadensis* at ca. 5 ka is the primary driver of the loss of ETS between 6 and 4 ka, while the gradual

recovery of *T. canadensis* after 5 ka drives the corresponding increase of ETS (Fig. 4). This shift in

dominance between summergreen and evergreen trees and shrubs, in turn, regulates the overall albedo of

the land surface and particularly its seasonal range. During times of foliage, summergreen forests can

have more than twice the albedo of evergreen forests (Hollinger et al., 2010). Summergreen forests also

exhibit much greater seasonal variability in albedo, with maximum values in full foliage being 20-50%

larger than annual lows.

Second, the peak in summergreen tree cover between ca. 8.5 and 4.5 ka (Fig. 2) may indicate a

seasonal-scale positive feedback loop between summer insolation, summergreen phenology, and summer

temperatures. Studies have suggested that the late-Pleistocene to early-Holocene peak in summer

insolation favored summergreen phenology and carbon acquisition strategies over evergreen strategies

(Delcourt and Delcourt, 1994; Williams and Jackson, 2007). Studies from the Pacific Northwest and

Alaska also show a peak in *Alnus* and other summergreen tree and shrub taxa during the early Holocene

(Fig. 7c), coincident with the summer insolation maximum and local maxima in temperature (Gavin et al.,

2013; Edwards et al., 2005; Whitlock, 1992). This climate-driven response, in turn, may have increased

the seasonal range of albedo and surface temperatures, thereby further favoring summergreen strategies.

This summergreen feedback might have also contributed to some summer cooling, due to the higher

summer albedo of summergreen trees (Hollinger et al., 2010). As an alternative hypothesis, Herzschuh,

(2020) suggested that the Eurasian distribution of evergreen spruce-dominated and deciduous larch-

dominated evergreen forests was due to historical contingencies and alternate stable states. Although

many studies have explored the effect of early to middle Holocene afforestation on vegetation feedbacks

in the northern latitudes (Brovkin et al., 2009; TEMPO (Testing Earth System Models with Paleo-

Observations), 1996), to our knowledge no paper has yet focused specifically on the seasonal feedback

effects associated with shifting proportions of summergreen and evergreen trees and shrubs.


**4.3 Uncertainties and limitations in pollen-vegetation model reconstructions**

Land cover reconstructions from pollen rely upon a variety of pollen-vegetation models (PVMs), some of

which have well-understood limitations and uncertainties, and some of which are newer and are still being

studied. The REVEALS PVM used here is has been widely adopted (e.g. Li et al., 2020; Githumbi et al.,





2022c; Serge et al., 2023; Azuara et al., 2019; Hoevers et al., 2022) because it represents some of the

taxon-level processes governing pollen production, transport, and representation (Sugita, 2007a; Prentice,

1985). However, processes such as atmospheric transport are simplified (Jackson and Lyford, 1999a) and

key parameters such as pollen productivity estimates (PPEs) carry uncertainties (Wieczorek and

Herzschuh, 2020; Broström et al., 2008; Hayashi et al., 2022) that translate to uncertainties in fractional-

weighted land cover area. Most PPE estimates are generated from models fitted to spatial networks of

vegetation surveys and surface pollen samples, and these PPE estimates depend strongly on models of

pollen dispersal (Theuerkauf et al., 2012).

Such issues may explain some of the surprising aspects of the reconstructions presented here. For

example, reconstructed *Acer* cover in the NEUS/SEC (Fig. 4c) is high compared to settlement-era

estimates from witness trees and land surveys (Thompson et al., 2013; Paciorek et al., 2016). Similarly,

reconstructed cover of *Abies* in both the NEUS/SEC and PCCS is higher than expected (Figs. 4c, 7c).

*Abies* and *Acer* are notoriously underrepresented in fossil pollen assemblages (Bradshaw and Webb,

1985), relative to independent surveys of tree abundance in the surrounding ecosystems, due to the low

pollen productivity of maple trees relative to other taxa (Finkelstein et al., 2006; Liu et al., 2022) and high

fall speeds of *Abies* (Jackson and Lyford, 1999b). Hence, a key value of process-based PVMs, such as

REVEALS, is the ability to correct for these known biases. However, REVEALS estimates are sensitive

to parameter choices and it is possible that the estimates of *Abies* and *Acer* are too high. There are few

estimates of *Abies* fall speeds compared to other taxa, and the fall speeds used here from Wieczorek and

Herzschuh (2020) are similar to the values for *Larix* that initial testing indicated led to *Larix*

overrepresentation.

REVEALS is not spatial or temporal in nature, so there is no representation of spatiotemporal

dependencies among site-level reconstructions. REVEALS estimates of uncertainty are based on the total

number of pollen grains counted in a sample and the error associated with the PPEs, but do not include

process uncertainty or uncertainty in other model inputs. The GMRF serves as a post-hoc interpolator to

generate spatio-temporally complete vegetation reconstructions, but this approach does not

mechanistically represent the underlying processes that link pollen to vegetation.

To address these limitations with the REVEALS workflow (spatio-temporal incompleteness,

input parameter uncertainty, and uncertainty quantification), other forms of PVMs have been developed.

ROPES uses pollen accumulation rates and REVEALS to estimate pollen productivity (Theuerkauf and

Couwenberg, 2018). However, the dependence of ROPES on pollen accumulation rates may limit its

widespread utility. At many sites, pollen accumulation rates have high uncertainties due to variations in

sedimentation rate, few radiometric dates and poor chronological controls, and spike counting

uncertainties (Perrotti et al., 2022). STEPPS is a Bayesian spatio-temporal PVM (Dawson et al., 2016,





2019b). While theoretically sound, STEPPS is computationally intensive and its estimates are dependent upon the density and quality of spatial forest compositional calibration datasets, limiting its applicability to regional-scale domains. Other spatial Bayesian models developed for high-resolution networks of
pollen and vegetation data suggest that if the spatial site density is too low, STEPPS and similarly structured PVMs can over-estimate pollen dispersal distance (Liu et al., 2022).

**4.4 Future work**

With North American REVEALS-based reconstructions now in place, all middle- and high-latitudes in the Northern Hemisphere have quantitative reconstructions of Holocene land cover that are well
constrained by dense networks of fossil pollen records (Githumbi et al., 2022a; Li et al., 2020). The reconstructions for North America and Europe also have estimates of uncertainty from the spatiotemporal
GMRF model (Pirzamanbein et al., 2018a). This hemispheric coverage now enables the creation of a next generation of modeling scenarios, well-constrained by data, to explore land cover feedbacks in the
Holocene climate system (Harrison et al., 2020). These land cover reconstructions also help identify places where new local-scale, site-based research are needed to better understand human impacts, better
constrain land cover changes in areas of sparse data and high uncertainty, and better understand how local and taxon-level dynamics scale up to affect regional- to continental-scale vegetation and climate
dynamics.

     This work also underscores the critical need for more work to better constrain PPEs and fall

speeds, particularly for North American plant taxa. The REVEALS estimates in this analysis appear to be particularly sensitive to the parameterizations for underrepresented taxa such as *Abies* and *Acer*.
Lastly, given the maturation of pollen-vegetation modeling as a field of study and the emergence of multiple PVMs, there is now the opportunity for intercomparison studies among PVMs and, perhaps,
the development of ensemble-based inferences of past vegetation cover. There is a long history of methodological comparisons in pollen-based paleoclimatic inferences and paleoclimatology more broadly
(Chevalier et al., 2020). Few systematic intercomparisons of PVMs yet exist (Roberts et al., 2018), although individual papers have discussed strengths and weaknesses of different modeling approaches
(Liu et al., 2022; Theuerkauf et al., 2012). Initial efforts are underway to compare land cover reconstructions from REVEALS-GMRF to those from the Bayesian spatio-temporal model STEPPS.
Moreover, given that ensemble-based approaches have been shown to have higher predictive ability in fields such as climate modeling and species distribution modeling (Deser et al., 2020; Bothe et al., 2013;
Rangel et al., 2009; Thuiller et al., 2009), there is value in developing approaches for ensemble-based PVMs for past land cover inference.



## 5. Conclusions

Continental-scale changes in land cover in North America during the Holocene are the outcome of multiple interacting drivers operating over a range of temporal and spatial scales. For much of the Holocene (ca. 8 to 1.5 ka), the spatial configurations of continental-scale fractional forest cover were broadly stable. Major continental-scale trends included early Holocene afforestation, a middle-Holocene peak in the areal proportion of summergreen trees and shrubs, and a last-millennium increase in open land. These trends were powered by taxon-level dynamics that varied within and among regions. The regional dynamics can be attributed to individualistic postglacial shifts in the range and abundance of plant taxa driven by changing climates and increased land area after deglaciation; abrupt and large species-level events, such as the mid-Holocene collapse of eastern hemlock; and a shift from localized land use during the late Holocene to massive ecosystem transformation following EuroAmerican settlement. This work rejects the ideas of both pre-industrial forest stability and widespread major conversions in land cover type.

This work contributes to the LandCover6k initiative to produce continental- to global-scale reconstructions of Holocene land cover dynamics that are well-constrained by proxy data and are quantified using consistent methods. These REVEALS-GMRF reconstructions can help refine Holocene models of land use and land cover change, provide a foundation for comparisons among PVMs, indicate priority areas for future new site-level research, and establish realistic benchmark scenarios constraining and assimilating with Earth system model simulations of the interactions among Holocene climate, land cover, and anthropogenic change. Scaling up from these continental to hemispheric and global products will enable further testing of hypotheses about the drivers and feedbacks of Holocene land cover change. These reconstructions also highlight several potentially important but unstudied vegetation-atmosphere feedbacks, including the collapse of eastern hemlock at ca. 5 ka and a positive feedback loop between mid-Holocene peaks in seasonality of insolation, temperature, and vegetation phenology.

Comparison of the REVEALS-GMRF land cover reconstructions with those inferred from alternate PVMs is a research priority. REVEALS-GMRF depends on a suite of input variables that are both uncertain and exhibit spatial and temporal variation not accounted for in the model. Also, the use of a post-hoc Bayesian GMRF interpolation step means that uncertainty of reconstructions associated with these assumptions is not fully characterized.

Finally, this work emphasizes the importance of quantifying ecosystem processes and feedbacks between the land surface and climate system. Holocene land cover changes in North America are non-negligible and there is a pressing need to refine ecosystem models, in particular when those models are coupled within an Earth System model. The availability of well-constrained data products has been a



major barrier to this effort. Paleoecology and palynology are reaching a new level of maturity with
respect to advances in data discovery and accessibility, as well as development of consistent analytical
methods that permit interpretability of large-scale spatio-temporal change. Data assimilation experiments
that integrate these reconstructions into Earth System models have the potential to resolve major
questions in the field, such as the Holocene Conundrum and disentangling the effects of climate-
vegetation-human interactions upon Holocene vegetation and climate dynamics.

## 6. Code and data availability

Code and data used in this workflow are publicly available at https://github.com/andydawson/reveals-na.
The REVEALS-GMRF reconstructions are available at
https://github.com/BehnazP/SpatioCompo_entireHolocene_NA. The description and implementation of the
digitization of lake area is available at https://github.com/NeotomaDB/neotoma_lakes.

## 7. Author contributions

AD and JWW co-designed and co-wrote the paper, with AD leading on REVEALS analyses and figure
generation. MJG helped initiate this effort, led PAGES LandCover6k, and coordinated activities with
other continental-scale mapping groups. BP and JL assisted with the GMRF modeling and analyses,
while SG assisted with Neotoma data provision and digitization of lake size. RSA, AB, DF, KG, DG, TL,
TM, WO, BS, and CW all contributed new records to the North American Pollen Database in Neotoma
and assisted in data interpretation. All authors contributed to manuscript development and revision.

## 8. Competing interests

The authors declare that they have no conflict of interest.

## 9. Acknowledgments

This work is a contribution to the Past Global Change (PAGES) project and its working group
LandCover6k (https://pastglobalchanges.org/science/wg/former/landcover6k/intro), which in turn
received support from the Swiss National Science Foundation, the Swiss Academy of Sciences, the US
National Science Foundation, and the Chinese Academy of Sciences. This work is also supported
through an NSERC Discovery Grant to A. Dawson. M.-J. Gaillard received support from the Swedish
Strategic Research Area (SRA) MERGE (ModElling the Regional and Global Earth system;



http://www.merge.lu.se.  Data were obtained from the Neotoma Paleoecology Database

(http://www.neotomadb.org) and its constituent database the North American Pollen Database. The work

of data contributors, data stewards, and the Neotoma community is gratefully acknowledged.  Support for

Neotoma is from the NSF-Geoinformatics program (1948926).  Digitization of basin areas was assisted

by Jenna Kilsevich, Grace Roper, Claire Rubbelke, Grace Tunski, Mathias Trachsel, and Bailey Zak.



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
