# Peer review of "Holocene land cover change in North America: continental"

_Climate of the Past, 2024_

## Referee Comment (RC1)

**Review of "Holocene land cover change in North America: continental trends, regional drivers, and implications for vegetation-atmosphere feedbacks"**

The manuscript by Dawson et al. presents new gridded reconstructions of land cover changes in North America, combining pollen-based vegetation cover reconstructions and a Bayesian spatial interpolation model. The new reconstructions are a valuable community effort and will be of great use for future studies of large-scale vegetation changes, land-atmosphere feedbacks, and anthropogenic land use during the Holocene. The maps can serve as boundary conditions for climate simulations and for evaluating Earth system model simulations with dynamic vegetation. Especially the high number of collected records covering the early Holocene is an impressive feature. The paper is well-written and my comments are mostly minor.

**General comments**

- The Bayesian interpolation methodology is sound and has been established over several studies. Nevertheless, it was originally developed for individual time slices (spatial reconstructions) while the new LandCover6k efforts and hopefully further data compilations in the future aim at spatio-temporal reconstructions. While I don't think that any adjustments of the interpolation strategy are needed for this study which does not aim at progressing the statistical interpolation methodology, I would appreciate discussing not just limitations of REVEALS but also of the interpolation methodology in Sect. 4.3. Moving from time slice to spatio-temporal reconstructions offers new statistical and data science challenges and opportunities which would be worthwhile discussing. In particular, can you comment on the potential for handling age uncertainties in the reconstruction algorithm, how uncertainties from REVEALS are propagated to the interpolation algorithm, using an actual spatio-temporal interpolation algorithm instead of reconstructing a set of time slices (see my comment below), and testing the impact of the non-uniform distribution of site locations through, e.g., bootstrapping. Do you think that cross-validation experiments in which some portion of the REVEALS reconstructions is left out from the interpolation could be a way to evaluate the spatial (or spatio-temporal) reconstructions?

- The GMRF method provides reconstruction uncertainties for all grid boxes. However, so far, the uncertainties are only visualized for the continental and regional mean curves. To understand the statistical significance of the spatial land cover variations, it would be very helpful to also plot maps of the reconstruction uncertainties, for example together with Fig. 2. For the change maps (Fig. 4a, 5a, 6a, 7a), hatching areas with statistically significant changes would be valuable to assess the importance of the temporal changes.

- There are three other aspects related to the applications and improvement of the datasets that I kindly ask the authors to discuss or enhance the respective discussion.
  The authors mention the under-representation of arid regions due to a lack of pollen records. Do you see prospects for including other proxies, either for vegetation or for (hydro-)climate to improve the reconstruction for arid regions (e.g., biomarkers, isotopes from speleothems, or lake levels)?
  Currently, only three land cover types are separated. Is there the potential in terms of data availability to also separate boreal, temperate, and subtropical forest / grassland types in addition to evergreen, summergreen, and open land?
  Finally, the current separation into time slices of 1kyr (and potentially further smoothing from

age uncertainties) precludes the analysis of sub-millennial trends. Do you see the possibility to also identify multi-decadal to multi-centennial variations on the regional and continental scale with the existing data coverage, potentially using an improved spatio-temporal interpolation methodology?

- The authors discuss implications for biophysical atmosphere-vegetation feedbacks very well (e.g., l.49-53, Sect. 4). In this context, it would be suitable in my opinion to also mention biogeochemical feedback mechanisms as the new land cover reconstructions should also be a useful tool for studying these ones, in particular since more and more Earth system models having capabilities for prognostic carbon and nutrient cycles.

- Data availability: The posterior mean reconstructions have been made available as csv files through a github repository. In the interest of maximing reusability and making it easy to cite the dataset, it would be very valuable to (i) publish the data sets also in a FAIR repository with a permanent identifier, and (ii) publish the reconstructions as netCDF files which are more suitable for gridded data and allow for a better interoperability with climate and vegetation simulations. Additionally, I would recommend to make not just the posterior means but also the uncertainties available in a suitable data format, either as marginal (point-wise) uncertainties or, better, by publishing MCMC samples which allow quantification of spatially correlated uncertainties.

**Specific comments:**

- Regarding all maps in the manuscript, please consider using a different projection that displays the size of regions better since in the current projection the high latitudes are heavily overrepresented compared to, e.g., Mexico.

- l. 33: I kindly ask you to use the term "Holocene temperature conundrum" instead of just "Holocene conundrum" given the number of other conundrums that have appeared in the literature over the last decade(s).

- l. 57: Would it be suitable to mention not just land cover dynamics in the last sentence of the abstract but also Holocene climate dynamics?

- l. 63: Should it be "net-negative" instead of "negative-net"?

- l. 114-117: Is there a specific reason why there was no prior continental-scale reconstruction for North America?

- l. 163: The authors use Bchron for the age modeling which is an appropriate choice in my opinion. I'm just curious if there is a specific reason to not use BACON which seems to be used more often in recent studies? While both model would be justifiable choices from my outsider view, it could be of interest to the community if the authors see specific advantages of Bchron for the vegetation reconstructions at hand.

- l. 168-169: Given that the taxa selection seems very important for the vegetation cover reconstructions, can you provide some more information on the selection criteria and the representativity of these taxa for the vegetation at the different locations? In addition, I would strongly suggest to move Supplementary Table 2 to the main manuscript (potentially with some additional information on the importance of the taxa like the average pollen percentage of those taxa).

- l. 192-207: Can you provide the number of lakes used in the different workflow steps?

- l. 218: Can you provide some support, e.g., in the supplement, for the very strong statement that "these vegetation reconstructions indisputably overrepresented larch"?

- l. 226: Please remove the gray shading in Trondman et al. (2015).

- l. 260-261: I don't understand the rationale behind the "mean relative cover". Excluding ice covered areas and areas without reliable reconstructions is reasonable, but why would you not use the area weighted mean of all grid boxes with reconstructed vegetation instead of just the average over those grid boxes?

- Fig. 1: To understand the spatio-temporal coverage properties better, can you provide maps similar to Fig. 1a for the individual time slices in the supplement?

- l. 332: I struggle to connect the mentioned increase from 56% to 91% with the results presented in Fig. 2 where the increase looks much smaller. Can you please clarify where these numbers come from?

- Fig. 4c: Spruce and pine are prominently mentioned in the text (l. 374-378) but are not included in the figure with the coverages of important taxa. Is there a reason for this exclusion?

- Sect. 4.1 and 4.2.1 are fairly long considering that they are mostly a literature review while being relative unconnected to results from the new reconstructions. Therefore, I'd ask you to either state more explicitly how the new results are in agreement with / contradicting previous studies or consider shortening this part given that the paper is already rather long. If the goal is mainly to state potential applications of the new reconstructions, I don't think this long discussion of the previous literature is needed.

- Sect. 4.1.1 - 4.1.3 should be 4.2.1 - 4.2.3.

- l. 784: It is stated that the GMRF creates a spatio-temporally complete vegetation reconstruction. Is this an appropriate characterization? From my understanding of the methods section and previous studies using the GMRF method, it creates spatially complete reconstructions for a set of predefined time slices but without considering temporal dependences between the time slices. If this is an misunderstanding on my site, it would be helpful to state more explicitly in the methods section how the time dimension is handled in the GMRF since the referenced studies only apply it for spatial reconstructions.

- l. 843-845: Maybe consider simplifying or splitting this sentence.

- l. 861: Is data assimilation the appropriate word here? From my understanding, the reconstructions would either be used as boundary conditions in simulations or for comparison with dynamically simulated vegetation, whereas data assimilation would refer to a dynamic simulation in which the simulated values would be relaxed towards the reconstructions. The latter is something that hasn't be done so far with vegetation reconstructions as far as I know.

- In many instances (e.g., l. 129, 196, 214, 216, 252), the parentheses in citations are set inconsistently. I kindly ask you to check the citations throughout the paper for typesetting errors.

- I kindly ask you to check the reference list again and harmonize the used reference style, in particular by providing DOIs consistently wherever available. Additionally, Dawson et al. 2019a and Dawson et al. 2019b seem to be the same reference, and Githumbi et al. 2022b and Githumbi et al. 2022c also seem to be the same reference.

---

## Author Comment (AC1)

**Review of "Holocene land cover change in North America: continental trends, regional drivers, and implications for vegetation-atmosphere feedbacks" by Andria Dawson, et al.**

This manuscript describes a study to reconstruct land cover for the Holocene over North America. As part of the LandCover6k initiative, the methodology follows a standardized procedure: first pollen records from sedimentary archives are synthesized and samples are assigned ages using up-to-date age-depth models. Then, pollen spectra are simplified and decimated to include specific taxa, and relative abundances of these taxa are passed to the REVEALS pollen-landscape model. REVEALS generates quantitative estimates of land cover for specific taxa that can be further generalized into broad groups of plant functional cover, e.g., broadleaf deciduous or needleleaf evergreen trees. These point-based data are then interpolated to a continuous 1-degree grid covering the study area. The work presented here complements similar activities undertaken for other parts of the Northern Hemisphere and ongoing work in the tropics and elsewhere. The authors present the results of the synthesis in the form of gridded maps and synthetic timeseries covering the entire North America spatial domain, and for specific regions that they analyze in further detail.

Overall, this is an excellent study that is rigorous in its methodology, interesting and in some ways novel in terms of results, and honest about shortcomings. The authors helpfully provide a roadmap for future research including on improving the land cover reconstructions and recommendations for research that could employ the maps and other datasets produced here. There are a few issues that should be clarified before publication, and ultimately this paper and the associated datasets will make a valuable contribution to the journal and support range of fields in further study.

We thank Reviewer 2 (Dr. Jed Kaplan) for these comments.  We consider ourselves fortunate to receive two detailed, thoughtful, and constructive reviews of our ms.

**General comments**

While changes in ice cover were considered, it appears that sea level changes (and proglacial lakes) were ignored in this study. This is a major limitation of the spatial analyses and at the very least should be justified. It's a bit strange because these paleogeographic changes are considered in previous, similar studies by some of the same authors (e.g., Williams, 2003; Williams et al., 2004). The early Holocene is characterized by very large proglacial lakes at the margin of the Laurentide Ice. More importantly were the postglacial isostatic adjustments that lasted throughout the Holocene. For example, the Hudson Bay Lowlands were submerged until after 5ka and low-lying areas of the Atlantic coast and Florida had significantly more land area exposed in the early Holocene. Data on sea level changes, for example from the PAGES PALSEA activity would be worth considering, and citing in an explanation of why these were not part of the current study.

Thank you for this comment.  We will add proglacial lakes to our mapping and areal analyses, since these lakes were still widespread in portions of the study during the early Holocene.  We

will cite recent reconstructions of sea level change (from PALSEA or other efforts) while noting that for the Holocene time period and continental-scale study presented here, these sea level effects are not expected to have a major effect on our reconstructions.

In the interpolated maps, the parts of the study domain that show no data I assume are because the "confidence region" (CR) was greater than the threshold of 9, for example in much of Mexico in the early Holocene. It would be helpful to see the CR maps themselves included among the supplementary figures. Looking at Figure 1, there are only 3 or maybe 4 sites in all of Mexico, so it is hard to understand, especially given the climatic and topographic diversity of Mexico, that there is much power in the interpolations over that space.
We agree that this would be helpful, and we will add maps of uncertainty for all time periods shown in the manuscript (see also response to Reviewer 1).

We will set a fixed domain size and exclude grid cells in the Mexico region in our analysis given the lack of records for this region. Setting a fixed domain size in Mexico removes confounding effects that may arise from the changing number of included grid cells.

All of the data products presented here (point-based and gridded maps) must be freely released on zenodo.org or other open-access data repository that provides a DOI upon final publication of the paper. The gridded maps should be provided in the earth system modeling-standard netCDF format.
We agree and will do this. Reviewer 1 raised the same point.

**A few notes on presentation**

As "land use" is generally accepted to be an activity that is unique to humans, it is not necessary to qualify the term with "human land use" or "anthropogenic land use" in the manuscript. In the interest of conciseness, please just use "land use" alone throughout the manuscript, or maybe define it once at the beginning of the text.
This is a good point, and we will edit the manuscript accordingly.

I found the constant switching back and forth between scientific names and common names for taxa distracting and sometimes confusing. Use of both nomenclatures even occurs in a single sentence (e.g., lines 435-436). I ask the authors to pick one nomenclature system and stick with it throughout the entire manuscript.
We will make this change.

Please use a thinner line thickness in all of the maps presented in the manuscript and supplement. The heavy line weight around the ice sheets and coastline distracts from the content. Perhaps the ice sheets could be plotted in a blue or brighter, contrasting color as polygons, without any outline at all.
We will revise the ice and coastline colors accordingly.

**Specific comments**

Lines 48-49
It is not at all clear how changes in the abundance of hemlock could have been significant enough to have a biogeophysical feedback to climate; see further comments below.
We address this point below.

Lines 168-169
Please explain briefly how relative abundances are calculated when some taxa are ignored? Is there an "all other taxa" bin? Or are only abundances relative to the considered taxa included? What happens when a taxon that is considered to be important in terms of land cover, even locally, is not part of those used in the REVEALS model?
We will address this in our revisions to the Methods section. The standard REVEALS workflow does not include an "all other taxa" bin. This is because of the variability in PPEs and fall speeds among taxa that would be included in such a bin. In this work, we first translated the Neotoma taxonomy to the Whitmore taxonomy (ref); this resulted in a list of 47 taxa. We identified the taxa that were most abundant and indicators of land cover type, of which there were 33. There were corresponding PPE and fallspeed values for all of these taxa. The set excluded results in a total of about 0.5% of the total pollen grains counted (for North American Holocene). We will add some text to clarify these decisions, and include a list of the taxa that were excluded in the supplement.

Line 235
Approximately how does the grid resolution of the 1x1 degree interpolated surface compare to the 10,000km2 area represented by a REVEALS reconstruction noted on line 179? Naturally it changes by latitude, but it would be helpful to put a comparative statement here.
We will add some text that discusses the area that REVEALS reconstructions represent, in the context of our grid cell size.

Lines 254-255
Here where CR is introduced, it would be good to call out supplementary figures here showing this value in map form for all periods.
As indicated above and in our response to Reviewer 1, we will add maps of reconstruction uncertainty to the supplementary information.

Lines 260-261
I understand that the LandCover6k grid was specified as 1x1 geographic degrees, but wouldn't it have made more sense to do the original work on an equal-area grid and then only reproject the data in a final step? At the very least it would have made interpretation of the maps more straightforward, and would be similar to earlier work (Williams, 2003; Williams et al., 2004).
Our primary goal was to maintain consistency with the other LandCover6k papers. We'll publicly share the results (as NetCDF files) in 1x1 degrees, but in our data visualizations, we'll reproject maps to Albers equal area, using the same projection parameters as in the earlier papers by Williams et al.

Figure 1
Could you plot the 1x1 degree graticule on this map using a very thin line in an unobtrusive color? It would make interpretation of the grid resolution of the other maps easier.
We'll experiment with adding this graticule, and will keep it as long as figure legibility is maintained.

Figure 1
Given the very high density of sites, it seems strange that nearly all of Minnesota is not included in any of the regional boxes. The choice to exclude this area deserves some explanation.
Our selection of regions was not intended to be comprehensive (this would have made for a very long and dull paper).  Rather, we picked selected regions, based on a) whether we saw an interesting trend in land cover to which we wanted to call the reader's attention and/or b) describing changes in the west, which has been less intensively studied.  Other regions certainly could have been chosen, and Minnesota definitely has its merits.  We will add a brief note about these criteria to the manuscript.

Figure 2
Use a thinner line weight, or no line at all for the ice sheet outline (as noted above).
We will revise the map color scheme accordingly.

Figure 2
To aid in quickly interpreting the plots and to provide better consistency with the rest of the figures, please plot the land cover fractional surfaces in the same colors as used in Fig. 3 and the other timeseries plots. That is to say, plot the first column of maps in shades of green, the second in shades of blue, and the third in shades of orange.
We recognize that this would likely make it easier to link the time series figures with the land cover maps, but one of the primary objectives of the map series is to compare land cover across the maps. Using different colors in land cover maps would make this difficult, and would require three legends instead of one. Given this, we would like to keep the single color scheme for the maps.

Figure 2
As noted in the supplement the three interpolated surfaces sum across to 100% in each row, and there is no "missing" fraction that represents bare ground. In the Arctic and in desert areas, the landscape is not 100% vegetated. This information should not be buried in the supplement, and needs to be clearly noted when the main figures are presented in the figure caption and body text. It should further be noted as a limitation and explained why this is not the case in the main manuscript text.
Agreed, we will clarify this in the main text.

Line 311
Given that there are only 3 sites in Mexico, is the spatial domain of the study justified? Wouldn't a maximum distance buffer around nearest site be better - e.g., up to 100 km apart (corresponding to the REVEALS indicative catchment area)? As noted above there is a distance

filter on the grid based on the CR value, but it would be interesting to see how this translates into distance from a site. Some statistics, such as the max distance from any site in the interpolation, would be helpful, even if only in the supplementary materials.

We will establish a standard spatial domain and reduce the size of the study domain to omit grid cells for which there are reconstructions only in a few time periods (e.g. Mexico, and perhaps several other coastal grid cells) .

Line 320
The number of gridcells contributing to the curves presented in Figure 3 changes based on ice area, apparently not sea level, but also CR value. Can we see an additional curve on this figure showing the total area in the spatial domain contributing to the cover estimate?

By establishing a standard domain size (see responses above), we will not need to include a curve representing changes in domain size over time.

Lines 435-436
In this sentence, and others, please just choose one form of plant nomenclature or the other, and stick with it.

Will do.

Lines 541-543
"… desert, steppe, and other open-land arid ecosystems are likely to be underrepresented in these reconstructions, due to a scarcity of dryland sites" yet the interpolated maps and timeseries curves imply continuous vegetation cover (without bare ground), if I understand correctly. This limitation of the methodology should be further described and justified.

We will make this edit.

Line 566
Is there really nothing to say here about sea level dynamics over the period?

We will add to this paragraph a brief mention of sea level changes over this time period.

Line 625
I suggest a small rewording of this sentence to: "During the late Holocene, the growth of Indigenous populations and intensification of land use in the Americas had increasing effects on land cover. Understanding the interactions among…"

We'll consider this rewording while reviewing this section to address, e.g., the next comment.

Line 632
Evidence for dense populations and land use in North America are dismissed here, yet a number of examples of this are provided in the following paragraph. This sentence could be reworded to better tie to what is coming next.

We'll look into ways to reword this sentence.

Paragraph starting on line 660

What is the purpose of this paragraph? Can it be tied back to the data presented in the current study?

We'll add a sentence or two that ties this paragraph back to the results.  The general goal of this paragraph is to at least briefly note the major effects of EuroAmerican land use, without going deep into this topic, as previous papers have covered this topic well.

Section starting on line 668 (4.1.1)

This section needs to be tied back more clearly to the findings in the current study, at least speculatively. The section reads like a review paper now and there is nothing new in here.

Per comments from Reviewer 1, we plan to keep this paragraph, while somewhat shortening its treatment of biogeophysical feedbacks and adding some text on carbon cycle feedbacks.   Part of the goal of this section is to broadly introduce the themes that follow; we'll look for ways to strengthen this connection.

Line 675-676

The full name of the "TEMPO" acronym could be removed here and just put in the bibliography.

Will do.

Paragraph starting on line 707

This paragraph does a very good job of explaining how the data synthesized in the current study ties back to previous work. It should be a model for how section 4.1.1 could be improved.

Thank you.

Line 715

"Great Plains"

We will fix this..

Line 728-731

It is not clear from this section or from the maps or timeseries how large, in absolute terms, the coverage of T. Canadensis could have ever been. The paragraph seems to insinuate that it could have been abundant enough to make a majority proportion of forest cover, therefore having a strong influence on, e.g., albedo. But… (see next comment)

See response to the next comment.

Lines 731-732

Am I missing something because I don't see a shift in the dominance in Fig. 4, which is always more than 50% summergreen trees and shrubs, with evergreen less than 30% cover fraction throughout the Holocene. Are you arguing that ETS forests were conifer-dominated? Otherwise, the albedo changes would have been very subtle, especially since T. Canadensis can persist in the understory for a century or longer and so while it is there and producing pollen, it will have no influence on summer albedo and relatively little on winter.

Yes, please note the interesting difference between Figure 4b and 4a.  The time series in Figure 4b show little change, as Reviewer 2 notes, but they are averaging across a broad area.  Figure 4a shows that there is a very large effect associated with the hemlock collapse, with 40%

changes in evergreen cover, but that these changes are concentrated in the eastern part of the study domain.  Hemlock is a late-successional shade-tolerant tree and tends to be a canopy dominant in areas of low disturbance.  We stand by our inference that this single-species collapse could have had a major effect on land-atmosphere interactions at regional to subregional scales, and perhaps more broadly, depending on how the teleconnections played out.

More broadly, this topic is a good example of how different phenomena are operating at different scales - one of the main points of this paper.

Lines 741-743
Looking at the summary figures, these changes must have all be very subtle. If not, then some further quantitative information should be highlighted here.
Based on the summary figures, the changes represented here are about 5% at a continental scale, which is worth reporting and discussing, because of the large spatial extent involved.   (A global mean temperature increase of 2C is a big deal; a 2C increase locally not as much.) This is another topic that is a good example of how different effects manifest at different scales.  The hemlock collapse was very large but at subregional to regional scales; this is a smaller change but across a much larger spatial domain.

Line 747
Broadleaf summergreen trees have greater maximum evapotranspiration rates than needleleaf evergreens and this effect should also be mentioned here as it is probably more important than the summertime albedo differences.
Agreed, we will mention this.

Line 753
I am not convinced that there is anything more than "relatively subtle shifts in the proportions of summergreen and evergreen trees and shrubs" shown in the data presented here.
See response above, for comment on L741-743.

Line 775-776
If "… REVEALS estimates are sensitive to parameter choices…" then why didn't you not just explore a larger parameter space and make a range of reconstructions? Instead of just one? Seems like it would be an easy change and could lead to the preparation of a range of maps or uncertainty fields.
In this manuscript, we are focusing on the REVEALS protocol and a careful review of the resulting reconstructions. We believe that adding a sensitivity analysis of REVEALS parameterizations is beyond the scope of this paper.

Line 784-785
The sentence mentions that "…this approach does not mechanistically represent the underlying processes that link pollen to vegetation". The GMRF method also does not account for soil, slope, aspect, and other edaphic controls on vegetation cover. This should be mentioned.

We will make this change.

Line 828-830
Here it is admitted that the changes in "… continental-scale fractional forest cover were broadly stable." This statement does not seem to support the idea that biogeophysical feedbacks between land and atmosphere would have been very important, in contrast to what is insinuated earlier in the manuscript. Some further explanation would be helpful here.
As noted above, different effects manifest at different scales, and this manuscript is designed to report phenomena across scales.  During revisions, we will review all statements and sections to ensure that they are clearly associated with the appropriate scale of inference.

References
Williams, J. W. (2003). Variations in tree cover in North America since the last glacial maximum. Global and Planetary Change, 35(1-2), 1-23. doi:10.1016/S0921-8181(02)00088-7
Williams, J. W., Shuman, B. N., Webb, T., Bartlein, P. J., & Leduc, P. L. (2004). Late-Quaternary Vegetation Dynamics in North America: Scaling from Taxa to Biomes. Ecological Monographs, 74(2), 309-334. doi:10.1890/02-4045

---

## Author Comment (AC2)

**Review of "Holocene land cover change in North America: continental trends, regional drivers, and implications for vegetation-atmosphere feedbacks"**

The manuscript by Dawson et al. presents new gridded reconstructions of land cover changes in North America, combining pollen-based vegetation cover reconstructions and a Bayesian spatial interpolation model. The new reconstructions are a valuable community effort and will be of great use for future studies of large-scale vegetation changes, land-atmosphere feedbacks, and anthropogenic land use during the Holocene. The maps can serve as boundary conditions for climate simulations and for evaluating Earth system model simulations with dynamic vegetation. Especially the high number of collected records covering the early Holocene is an impressive feature. The paper is well-written and my comments are mostly minor.

We thank Reviewer 1 for their positive and constructive review, and for their thoughtful and detailed comments.

**General comments**

1. The Bayesian interpolation methodology is sound and has been established over several studies. Nevertheless, it was originally developed for individual time slices (spatial reconstructions) while the new LandCover6k efforts and hopefully further data compilations in the future aim at spatio-temporal reconstructions. While I don't think that any adjustments of the interpolation strategy are needed for this study which does not aim at progressing the statistical interpolation methodology, I would appreciate discussing not just limitations of REVEALS but also of the interpolation methodology in Sect. 4.3. Moving from time slice to spatio-temporal reconstructions offers new statistical and data science challenges and opportunities which would be worthwhile discussing. In particular, can you comment on the potential for handling age uncertainties in the reconstruction algorithm, how uncertainties from REVEALS are propagated to the interpolation algorithm, using an actual spatio-temporal interpolation algorithm instead of reconstructing a set of time slices (see my comment below), and testing the impact of the non-uniform distribution of site locations through, e.g., bootstrapping. Do you think that cross-validation experiments in which some portion of the REVEALS reconstructions is left out from the interpolation could be a way to evaluate the spatial (or spatio-temporal) reconstructions?

   The REVEALS-GMRF interpolation approach used here is consistent with the approach used in the land cover reconstructions for Europe and China (e.g., Githumbi et al., 2022a and Li et al., 2023, respectively) . The process operates on individual time slices, and the interpolation approach itself does not have a temporal component. We will clarify this in the text as needed. We are in the process of developing an approach that includes a temporal component; the challenge is adding this complexity in a way that allows the approach to still be computationally tractable. We recognize that sample age uncertainty may influence results. The methods used in this work do not account for this age

uncertainty. Given the temporal grain of the time bins (500 years throughout the Holocene, except for the last 700 years, 100-350 years), we expect that within uncertainty the majority of sample ages will not shift among time bins. When a temporal component is added to the REVEALS-GMRF approach, we will consider how to account for this uncertainty.

REVEALS does quantify standard errors associated with relative abundance estimates. These standard errors are not considered by the REVEALS-GMRF approach. This means that uncertainty estimates from the REVEALS-GMRF approach quantify the uncertainty determined by the variability of REVEALS fractional land cover around the estimated spatial field.

We agree that validation experiments would be useful to understand the impacts of sample unevenness. Validation experiments have been done using the REVEALS-GMRF approach for Europe (Pizamanbein et al., 2018). In that effort a 6-fold cross validation technique where 10, 3 and 1 random selections of the block were left out. This work generally showed little change in the predictions when data was withheld.

The challenge in repeating these experiments for North America lies in the computational burden associated with the REVEALS-GMRF approach. Future work aims to improve computational efficiency of the approach, at which point such experiments will be possible.

We will add further discussion of uncertainty handling in the main text.

2. The GMRF method provides reconstruction uncertainties for all grid boxes. However, so far, the uncertainties are only visualized for the continental and regional mean curves. To understand the statistical significance of the spatial land cover variations, it would be very helpful to also plot maps of the reconstruction uncertainties, for example together with Fig. 2. For the change maps (Fig. 4a, 5a, 6a, 7a), hatching areas with statistically significant changes would be valuable to assess the importance of the temporal changes.
To address this comment, we will add maps of the reconstruction uncertainties to the Supplementary Information.

3. There are three other aspects related to the applications and improvement of the datasets that I kindly ask the authors to discuss or enhance the respective discussion. The authors mention the under-representation of arid regions due to a lack of pollen records. Do you see prospects for including other proxies, either for vegetation or for (hydro-)climate to improve the reconstruction for arid regions (e.g., biomarkers, isotopes from speleothems, or lake levels)? Currently, only three land cover types are separated. Is there the potential in terms of data availability to also separate boreal, temperate, and subtropical forest / grassland types in addition to evergreen, summergreen, and open land? Finally, the current separation into time slices of 1kyr (and potentially further

smoothing from 1 age uncertainties) precludes the analysis of sub-millennial trends. Do you see the possibility to also identify multi-decadal to multi-centennial variations on the regional and continental scale with the existing data coverage, potentially using an improved spatio-temporal interpolation methodology?

These are all excellent points and we plan to add a short review of them to the discussion, when we discuss future work and next steps. First, we agree that other proxies could be used in arid regions to improve vegetation reconstructions. Second, we agree that it would be possible to separate the land cover types into boreal, temperate, and subtropical components, given what is known about the individual pollen types and their climatic affinities. However, statistically modeling more finely resolved land cover groupings or taxa is complicated by the many 0 count observations in more finely resolved groupings, especially for such a large spatio-temporal domain. Third, whether centennial-scale variations can be confidently interpolated to a continental extent is a more open question, given the relative scarcity of sufficiently well-sampled and well-dated records. However, it should be possible to study multi-decadal to multi-centennial variations in vegetation cover at regional to continental scales, by carefully selecting records with the highest sampling resolution and age constraints.

4. The authors discuss implications for biophysical atmosphere-vegetation feedbacks very well (e.g., l.49-53, Sect. 4). In this context, it would be suitable in my opinion to also mention biogeochemical feedback mechanisms as the new land cover reconstructions should also be a useful tool for studying these ones, in particular since more and more Earth system models having capabilities for prognostic carbon and nutrient cycles.

We plan to add a short section reviewing biogeochemical feedback mechanisms, with an emphasis on carbon cycle implications.

5. Data availability: The posterior mean reconstructions have been made available as csv files through a github repository. In the interest of maximing reusability and making it easy to cite the dataset, it would be very valuable to (i) publish the data sets also in a FAIR repository with a permanent identifier, and (ii) publish the reconstructions as netCDF files which are more suitable for gridded data and allow for a better interoperability with climate and vegetation simulations. Additionally, I would recommend to make not just the posterior means but also the uncertainties available in a suitable data format, either as marginal (point-wise) uncertainties or, better, by publishing MCMC samples which allow quantification of spatially correlated uncertainties.

We will make the reconstructions publicly available in a NetCDF format in a FAIR-compliant repository such as PANGAEA or DRYAD, for both the means and uncertainties. Reviewer 2 raised the same point.

**Specific comments:**

Regarding all maps in the manuscript, please consider using a different projection that displays the size of regions better since in the current projection the high latitudes are heavily overrepresented compared to, e.g., Mexico.

We will change the projection to Albers equal area with standard parallels 33.333N and 66.667N and center point of 57N, 100W.

l. 33: I kindly ask you to use the term "Holocene temperature conundrum" instead of just "Holocene conundrum" given the number of other conundrums that have appeared in the literature over the last decade(s).
Will do.

l. 57: Would it be suitable to mention not just land cover dynamics in the last sentence of the abstract but also Holocene climate dynamics?
Agreed, will do.

l. 63: Should it be "net-negative" instead of "negative-net"?
Will do.

l. 114-117: Is there a specific reason why there was no prior continental-scale reconstruction for North America?
We will add a bit more historical context.

l. 163: The authors use Bchron for the age modeling which is an appropriate choice in my opinion. I'm just curious if there is a specific reason to not use BACON which seems to be used more often in recent studies? While both model would be justifiable choices from my outsider view, it could be of interest to the community if the authors see specific advantages of Bchron for the vegetation reconstructions at hand.
Both BACON and Bchron are widely used and follow very similar Bayesian modeling frameworks.  We have used both in our prior work.  We have found that Bchron establishes increasing uncertainty in age estimates between control points and requires fewer assumptions about sedimentary processes than Bacon. We will add some text (and possibly a supplementary figure) to support this decision.

l. 168-169: Given that the taxa selection seems very important for the vegetation cover reconstructions, can you provide some more information on the selection criteria and the representativity of these taxa for the vegetation at the different locations? In addition, I would strongly suggest to move Supplementary Table 2 to the main manuscript (potentially with some additional information on the importance of the taxa like the average pollen percentage of those taxa).
We will add information about why we chose the selected taxa.  Much of the pollen-parameter information in Supplementary Table 2 has been published elsewhere, so we tend to think that it better belongs in the Supplementary Information, but we can promote it to the main text if the Editor so requests.

l. 192-207: Can you provide the number of lakes used in the different workflow steps?
Yes, we will add this.

l. 218: Can you provide some support, e.g., in the supplement, for the very strong statement that "these vegetation reconstructions indisputably overrepresented larch"?

We will remove the word 'indisputably' from the revised ms. Larch is known to be a challenging taxon to model accurately in pollen-vegetation models, because it prefers wetland settings (at least towards the south of its range) and its large pollen grain does not disperse far and preserve well. Hence, there is a general challenge of differentiating the local populations of larch growing at the site where a core was retrieved from those occurring across the broader source area. And the scarcity of larch pollen grains means that the REVEALS reconstructions for larch are highly sensitive to the pollen-parameter settings for larch (i.e. PPEs and fall speed). When revising the text, we will add these points to either the main text or supplement.

l. 226: Please remove the gray shading in Trondman et al. (2015). 2

Will do.

l. 260-261: I don't understand the rationale behind the "mean relative cover". Excluding ice covered areas and areas without reliable reconstructions is reasonable, but why would you not use the area weighted mean of all grid boxes with reconstructed vegetation instead of just the average over those grid boxes?

There are two different effects that we are trying to capture with these two metrics. If we only do the area-weighted mean in which the unglaciated land area for a given time period is the denominator (and this is one of our two metrics), then this metric helpfully describes the proportional mix for any given time period, but will not capture the total increase in vegetated land area across time intervals, as North America deglaciates. In the second vegetation metric, we set the denominator to the total deglaciated land area at 0.25 ka. This second metric, by using a constant denominator that represents late-Holocene deglaciated land area, captures the general increase in vegetated land areas and of individual components.

The manuscript text on this topic could be clearer and we will work to revise it; we will also develop better and more precise terminology for these two metrics.

Fig. 1: To understand the spatio-temporal coverage properties better, can you provide maps similar to Fig. 1a for the individual time slices in the supplement?

Yes, we will add maps of site coverage for individual time slices to the supplement.

l. 332: I struggle to connect the mentioned increase from 56% to 91% with the results presented in Fig. 2 where the increase looks much smaller. Can you please clarify where these numbers come from?

This is an error in the text, caused by a failure to update results from an earlier version. Thank you for catching the mistake. We will fix the text.

Fig. 4c: Spruce and pine are prominently mentioned in the text (l. 374-378) but are not included in the figure with the coverages of important taxa. Is there a reason for this exclusion?

We will revise Fig. 4c to add spruce and pine, but may need to make further adjustments to preserve figure readability and information density.

Sect. 4.1 and 4.2.1 are fairly long considering that they are mostly a literature review while being relative unconnected to results from the new reconstructions. Therefore, I'd ask you to either state more explicitly how the new results are in agreement with / contradicting previous studies or consider shortening this part given that the paper is already rather long. If the goal is mainly to state potential applications of the new reconstructions, I don't think this long discussion of the previous literature is needed.

We will look for ways to shorten this section and better connect to results. We may move some of this to Discussion.

Sect. 4.1.1 - 4.1.3 should be 4.2.1 - 4.2.3.
We will fix this.

l. 784: It is stated that the GMRF creates a spatio-temporally complete vegetation reconstruction. Is this an appropriate characterization? From my understanding of the methods section and previous studies using the GMRF method, it creates spatially complete reconstructions for a set of predefined time slices but without considering temporal dependences between the time slices. If this is an misunderstanding on my site, it would be helpful to state more explicitly in the methods section how the time dimension is handled in the GMRF since the referenced studies only apply it for spatial reconstructions.

You are correct with your understanding of our methods. We will amend the text to clarify this point.

l. 843-845: Maybe consider simplifying or splitting this sentence.
Will do.

l. 861: Is data assimilation the appropriate word here? From my understanding, the reconstructions would either be used as boundary conditions in simulations or for comparison with dynamically simulated vegetation, whereas data assimilation would refer to a dynamic simulation in which the simulated values would be relaxed towards the reconstructions. The latter is something that hasn't be done so far with vegetation reconstructions as far as I know.

The word choice of data assimilation was intentional and the goal was to point to future research directions. Agreed that no data assimilation efforts have yet been done with pollen-based vegetation reconstructions and Earth system models, but we view this as a future potential application of this vegetation reconstruction and identical/similar ones from other continents (existing and to come).

In many instances (e.g., l. 129, 196, 214, 216, 252), the parentheses in citations are set inconsistently. I kindly ask you to check the citations throughout the paper for typesetting errors.
Will do and apologies for these formatting errors.

I kindly ask you to check the reference list again and harmonize the used reference style, in particular by providing DOIs consistently wherever available. Additionally, Dawson et al. 2019a

and Dawson et al. 2019b seem to be the same reference, and Githumbi et al. 2022b and Githumbi et al. 2022c also seem to be the same reference. 3

Will do.  We'll both track down DOIs and remove these duplicates.

---

## Author Response (AR3)

**Reviewer 1 Comments & Responses**

Review of "Holocene land cover change in North America: continental trends, regional drivers, and implications for vegetation-atmosphere feedbacks"

The manuscript by Dawson et al. presents new gridded reconstructions of land cover changes in North America, combining pollen-based vegetation cover reconstructions and a Bayesian spatial interpolation model. The new reconstructions are a valuable community effort and will be of great use for future studies of large-scale vegetation changes, land-atmosphere feedbacks, and anthropogenic land use during the Holocene. The maps can serve as boundary conditions for climate simulations and for evaluating Earth system model simulations with dynamic vegetation. Especially the high number of collected records covering the early Holocene is an impressive feature. The paper is well-written and my comments are mostly minor. We thank Reviewer 1 for their positive and constructive review, and for their thoughtful and detailed comments.

**General comments**

1. The Bayesian interpolation methodology is sound and has been established over several studies. Nevertheless, it was originally developed for individual time slices (spatial reconstructions) while the new LandCover6k efforts and hopefully further data compilations in the future aim at spatio-temporal reconstructions. While I don't think that any adjustments of the interpolation strategy are needed for this study which does not aim at progressing the statistical interpolation methodology, I would appreciate discussing not just limitations of REVEALS but also of the interpolation methodology in Sect. 4.3. Moving from time slice to spatio-temporal reconstructions offers new statistical and data science challenges and opportunities which would be worthwhile discussing. In particular, can you comment on the potential for handling age uncertainties in the reconstruction algorithm, how uncertainties from REVEALS are propagated to the interpolation algorithm, using an actual spatio-temporal interpolation algorithm instead of reconstructing a set of time slices (see my comment below), and testing the impact of the non-uniform distribution of site locations through, e.g., bootstrapping. Do you think that cross-validation experiments in which some portion of the REVEALS reconstructions is left out from the interpolation could be a way to evaluate the spatial (or spatio-temporal) reconstructions?

We have added further discussion of uncertainty handling in the main text.

The REVEALS-GMRF interpolation approach used here is consistent with the approach used in the land cover reconstructions for Europe and China (e.g., Githumbi et al., 2022a and Li et al., 2023, respectively). The process operates on individual time slices, and the

interpolation approach itself does not have a temporal component. This has been clarified throughout the text. We are in the process of developing an approach that includes a temporal component; the challenge is adding this complexity in a way that allows the approach to still be computationally tractable. We recognize that sample age uncertainty may influence results. The methods used in this work do not account for this age uncertainty. Given the temporal grain of the time bins (500 years throughout the Holocene, except for the last 700 years, 100-350 years), we expect that within uncertainty the majority of sample ages will not shift among time bins. In future work, we hope to add a temporal component to the REVEALS-GMRF approach; in this case we hope to consider how to account for this uncertainty. We have added text discussing the age uncertainty.

REVEALS does quantify standard errors associated with relative abundance estimates. These standard errors are not considered by the REVEALS-GMRF approach. This means that uncertainty estimates from the REVEALS-GMRF approach quantify the uncertainty determined by the variability of REVEALS fractional land cover around the estimated spatial field. We have clarified this in the text.

We agree that validation experiments would be useful to understand the impacts of sample unevenness. Validation experiments have been done using the REVEALS-GMRF approach for Europe (Pizamanbein et al., 2018). In that effort a 6-fold cross validation technique where 10, 3 and 1 random selections of the block were left out. This work generally showed little change in the predictions when data was withheld.

The challenge in repeating these experiments for North America lies in the computational burden associated with the REVEALS-GMRF approach. Future work aims to improve computational efficiency of the approach, at which point such experiments will be possible. We have added text discussing cross-validation experiments.

2. The GMRF method provides reconstruction uncertainties for all grid boxes. However, so far, the uncertainties are only visualized for the continental and regional mean curves. To understand the statistical significance of the spatial land cover variations, it would be very helpful to also plot maps of the reconstruction uncertainties, for example together with Fig. 2. For the change maps (Fig. 4a, 5a, 6a, 7a), hatching areas with statistically significant changes would be valuable to assess the importance of the temporal changes.

To address this comment, we have added maps of the reconstruction uncertainties to the Supplementary Information. We considered adding hatching, but ultimately decided against this given that the GMRF interpolation approach does not account for uncertainty in the REVEALS reconstructions (which in turn do not appropriately characterize uncertainty). We plan to address this question about uncertainty quantification of reconstructions in future work.

3. There are three other aspects related to the applications and improvement of the datasets that I kindly ask the authors to discuss or enhance the respective discussion. The authors mention the under-representation of arid regions due to a lack of pollen records. Do you see prospects for including other proxies, either for vegetation or for (hydro-)climate to improve the reconstruction for arid regions (e.g., biomarkers, isotopes from speleothems, or lake levels)? Currently, only three land cover types are separated. Is there the potential in terms of data availability to also separate boreal, temperate, and subtropical forest / grassland types in addition to evergreen, summergreen, and open land? Finally, the current separation into time slices of 1kyr (and potentially further smoothing from 1 age uncertainties) precludes the analysis of sub-millennial trends. Do you see the possibility to also identify multi-decadal to multi-centennial variations on the regional and continental scale with the existing data coverage, potentially using an improved spatio-temporal interpolation methodology?

These are all excellent points and have added a short review of them to Section 4.4 of the discussion, when we discuss future work and next steps. First, we agree that other proxies could be used in arid regions to improve vegetation reconstructions. Second, we agree that it would be possible to separate the land cover types into boreal, temperate, and subtropical components, given what is known about the individual pollen types and their climatic affinities. However, statistically modeling more finely resolved land cover groupings or taxa is complicated by the many 0 count observations in more finely resolved groupings, especially for such a large spatio-temporal domain. Third, whether centennial-scale variations can be confidently interpolated to a continental extent is a more open question, given the relative scarcity of sufficiently well-sampled and well-dated records. However, it should be possible to study multi-decadal to multi-centennial variations in vegetation cover at regional to continental scales, by carefully selecting records with the highest sampling resolution and age constraints.

4. The authors discuss implications for biophysical atmosphere-vegetation feedbacks very well (e.g., I.49-53, Sect. 4). In this context, it would be suitable in my opinion to also mention biogeochemical feedback mechanisms as the new land cover reconstructions should also be a useful tool for studying these ones, in particular since more and more Earth system models having capabilities for prognostic carbon and nutrient cycles.

We have retitled section 4.2.1 to "Biogeophysical and biogeochemical vegetation-atmosphere feedbacks to Holocene climates" and expanded it by adding a paragraph reviewing the carbon cycle literature. We have also added mention of biogeochemical feedbacks to the Abstract, Introduction, and Conclusion.

5. Data availability: The posterior mean reconstructions have been made available as csv files through a github repository. In the interest of maximing reusability and making it easy to cite the dataset, it would be very valuable to (i) publish the data sets also in a FAIR repository with a permanent identifier, and (ii) publish the reconstructions as netCDF files which are more suitable for gridded data and allow for a better interoperability with climate and vegetation simulations. Additionally, I would recommend

to make not just the posterior means but also the uncertainties available in a suitable data format, either as marginal (point-wise) uncertainties or, better, by publishing MCMC samples which allow quantification of spatially correlated uncertainties.

We have archived the North American Holocene land cover reconstructions from both REVEALS and REVEALS-GMRF on Dryad: <a href="https://doi.org/10.5061/dryad.c2fgz61m5">https://doi.org/10.5061/dryad.c2fgz61m5</a>.

**Specific comments:**

Regarding all maps in the manuscript, please consider using a different projection that displays the size of regions better since in the current projection the high latitudes are heavily overrepresented compared to, e.g., Mexico.

We have changed the projection used for map presentation to Albers equal area with standard parallels 33.333N and 66.667N and center point of 57N, 100W. However, all analysis was done using the WGS1984 standard, to be consistent with the Landcover6k northern hemisphere synthesis.

I. 33: I kindly ask you to use the term "Holocene temperature conundrum" instead of just "Holocene conundrum" given the number of other conundrums that have appeared in the literature over the last decade(s).

We have made this change throughout the manuscript.

I. 57: Would it be suitable to mention not just land cover dynamics in the last sentence of the abstract but also Holocene climate dynamics?

Done, and also added a mention of carbon cycle dynamics: "making it possible to better understand the regional- to global-scale processes driving Holocene land- cover, carbon-cycle, and climate dynamics."

I. 63: Should it be "net-negative" instead of "negative-net"?

**Fixed.**

I. 114-117: Is there a specific reason why there was no prior continental-scale reconstruction for North America?

No real scientific reason... mostly because REVEALS was more quickly and strongly adopted by the European community and the North American community has been working on other projects. We've left the main text here unchanged.

I. 163: The authors use Bchron for the age modeling which is an appropriate choice in my opinion. I'm just curious if there is a specific reason to not use BACON which seems to be used more often in recent studies? While both model would be justifiable choices from my outsider

view, it could be of interest to the community if the authors see specific advantages of Bchron for the vegetation reconstructions at hand.

Both BACON and Bchron are widely used and follow very similar Bayesian modeling frameworks. We have used both in our prior work. The differences are subtle, but we have found that Bchron establishes increasing uncertainty in age estimates between control points and requires fewer assumptions about sedimentary processes than Bacon. We experimented with adding text to describe this point, but the distinctions are too nuanced to explain briefly, and a fuller treatment felt like too much of a distraction from the main points of the paper, so ultimately we opted to not make this change. We did add text in 2.1 referring readers to literature that describes and evaluates the widely used age-depth model algorithms.

I. 168-169: Given that the taxa selection seems very important for the vegetation cover reconstructions, can you provide some more information on the selection criteria and the representativity of these taxa for the vegetation at the different locations? In addition, I would strongly suggest to move Supplementary Table 2 to the main manuscript (potentially with some additional information on the importance of the taxa like the average pollen percentage of those taxa).

We have added information about taxon selection as well as a list of excluded taxa. Much of the pollen-parameter information in Supplementary Table 2 has been published elsewhere, so we believe that it better belongs in the Supplementary Information, where we have left it. However, we can promote it to the main text if the Editor so requests.

I. 192-207: Can you provide the number of lakes used in the different workflow steps? Yes, we have added the number of lakes whose areas we were able to determine, and the number of lakes we assigned the standard medium size.

I. 218: Can you provide some support, e.g., in the supplement, for the very strong statement that "these vegetation reconstructions indisputably overrepresented larch"?

We have removed the word 'indisputably' from the ms. Larch is known to be a challenging taxon to model accurately in pollen-vegetation models, because it prefers wetland settings (at least towards the south of its range) and its large pollen grain does not disperse far and preserve well. Hence, there is a general challenge of differentiating the local populations of larch growing at the site where a core was retrieved from those occurring across the broader source area. And the scarcity of larch pollen grains means that the REVEALS reconstructions for larch are highly sensitive to the pollen-parameter settings for larch (i.e. PPEs and fall speed). We have added a version of the above text to the Discussion in Section 4.3, Uncertainties and limitations.

I. 226: Please remove the gray shading in Trondman et al. (2015). 2

We have removed the shading.

I. 260-261: I don't understand the rationale behind the "mean relative cover". Excluding ice covered areas and areas without reliable reconstructions is reasonable, but why would you not use the area weighted mean of all grid boxes with reconstructed vegetation instead of just the average over those grid boxes?

There are two different effects that we are trying to capture with these two metrics. If we only do the area-weighted mean in which the unglaciated land area for a given time period is the denominator (and this is one of our two metrics), then this metric helpfully describes the proportional mix for any given time period, but will not capture the total increase in vegetated land area across time intervals, as North America deglaciates. In the second vegetation metric, we set the denominator to the total deglaciated land area at 0.25 ka. This second metric, by using a constant denominator that represents late-Holocene deglaciated land area, captures the general increase in vegetated land areas and of individual components.

We have revised section 2.4 to clarify the language; we have also developed better and more precise terminology for these two metrics. We now use "relative-to-t" for the first metric and "relative-to-modern" for the second metric.

Fig. 1: To understand the spatio-temporal coverage properties better, can you provide maps similar to Fig. 1a for the individual time slices in the supplement?

We have added maps of site coverage for individual time slices to the supplement (Supp. Fig. S1).

I. 332: I struggle to connect the mentioned increase from 56% to 91% with the results presented in Fig. 2 where the increase looks much smaller. Can you please clarify where these numbers come from?

This is an error in the text, caused by a failure to update results from an earlier version. Thank you for catching the mistake. We have fixed this in the text, to indicate the increase from 42 to 61%.

Fig. 4c: Spruce and pine are prominently mentioned in the text (I. 374-378) but are not included in the figure with the coverages of important taxa. Is there a reason for this exclusion? We have revised Fig. 4c to include spruce and pine.

Sect. 4.1 and 4.2.1 are fairly long considering that they are mostly a literature review while being relative unconnected to results from the new reconstructions. Therefore, I'd ask you to either state more explicitly how the new results are in agreement with / contradicting previous studies or consider shortening this part given that the paper is already rather long. If the goal is mainly to state potential applications of the new reconstructions, I don't think this long discussion of the previous literature is needed.

We have modified Section 4.1 to better connect to results, by adding several figure pointers that better connect this discussion of drivers to the patterns of vegetation change reported in the results. In our view, this section is critical for giving context and potential explanations for the observed vegetation changes. The question of human activity vs. climate change as drivers of Holocene vegetation change is particularly sensitive and complex and so we have sought to provide a careful and nuanced discussion. One of the values of this paper is not just the data presented but also the expertise of the author team; this section summarizes this expert consensus.

We shortened the paragraph in section 4.2.1 that focused on biogeophysical feedbacks and added a paragraph that discusses biogeochemical feedbacks, per a comment above.

Sect. 4.1.1 - 4.1.3 should be 4.2.1 - 4.2.3.

**This has been fixed.**

I. 784: It is stated that the GMRF creates a spatio-temporally complete vegetation reconstruction. Is this an appropriate characterization? From my understanding of the methods section and previous studies using the GMRF method, it creates spatially complete reconstructions for a set of predefined time slices but without considering temporal dependences between the time slices. If this is an misunderstanding on my site, it would be helpful to state more explicitly in the methods section how the time dimension is handled in the GMRF since the referenced studies only apply it for spatial reconstructions.

You are correct with your understanding of our methods. We have added text to section 4.2 that clarifies this and discusses uncertainty related to REVEALS and REVEALS-GMRF.

I. 843-845: Maybe consider simplifying or splitting this sentence.

We have split the sentence to improve readability.

I. 861: Is data assimilation the appropriate word here? From my understanding, the reconstructions would either be used as boundary conditions in simulations or for comparison with dynamically simulated vegetation, whereas data assimilation would refer to a dynamic simulation in which the simulated values would be relaxed towards the reconstructions. The latter is something that hasn't be done so far with vegetation reconstructions as far as I know.

The word choice of data assimilation was intentional and the goal was to point to future research directions. Agreed that no data assimilation efforts have yet been done with pollen-based vegetation reconstructions and Earth system models, but we view this as a future potential application of this vegetation reconstruction and identical/similar ones from other continents (existing and to come).

In many instances (e.g., I. 129, 196, 214, 216, 252), the parentheses in citations are set inconsistently. I kindly ask you to check the citations throughout the paper for typesetting errors.

We have addressed this (and apologize for these formatting errors in initial submission).

I kindly ask you to check the reference list again and harmonize the used reference style, in particular by providing DOIs consistently wherever available. Additionally, Dawson et al. 2019a and Dawson et al. 2019b seem to be the same reference, and Githumbi et al. 2022b and Githumbi et al. 2022c also seem to be the same reference.

We've tracked down DOIs and removed these duplicates. We also double-checked journal titles for consistent capitalization.

**Reviewer 2 Comments & Responses**

Review of "Holocene land cover change in North America: continental trends, regional drivers, and implications for vegetation-atmosphere feedbacks" by Andria Dawson, et al.

This manuscript describes a study to reconstruct land cover for the Holocene over North America. As part of the LandCover6k initiative, the methodology follows a standardized procedure: first pollen records from sedimentary archives are synthesized and samples are assigned ages using up-to-date age-depth models. Then, pollen spectra are simplified and decimated to include specific taxa, and relative abundances of these taxa are passed to the REVEALS pollen-landscape model. REVEALS generates quantitative estimates of land cover for specific taxa that can be further generalized into broad groups of plant functional cover, e.g., broadleaf deciduous or needleleaf evergreen trees. These point-based data are then interpolated to a continuous 1-degree grid covering the study area. The work presented here complements similar activities undertaken for other parts of the Northern Hemisphere and ongoing work in the tropics and elsewhere. The authors present the results of the synthesis in the form of gridded maps and synthetic timeseries covering the entire North America spatial domain, and for specific regions that they analyze in further detail.

Overall, this is an excellent study that is rigorous in its methodology, interesting and in some ways novel in terms of results, and honest about shortcomings. The authors helpfully provide a roadmap for future research including on improving the land cover reconstructions and recommendations for research that could employ the maps and other datasets produced here. There are a few issues that should be clarified before publication, and ultimately this paper and the associated datasets will make a valuable contribution to the journal and support range of fields in further study.

We thank Reviewer 2 (Dr. Jed Kaplan) for these comments. We consider ourselves fortunate to receive two detailed, thoughtful, and constructive reviews of our ms.

**General comments**

While changes in ice cover were considered, it appears that sea level changes (and proglacial lakes) were ignored in this study. This is a major limitation of the spatial analyses and at the very least should be justified. It's a bit strange because these paleogeographic changes are considered in previous, similar studies by some of the same authors (e.g., Williams, 2003; Williams et al., 2004). The early Holocene is characterized by very large proglacial lakes at the margin of the Laurentide Ice. More importantly were the postglacial isostatic adjustments that lasted throughout the Holocene. For example, the Hudson Bay Lowlands were submerged until after 5ka and low-lying areas of the Atlantic coast and Florida had significantly more land area exposed in the early Holocene. Data on sea level changes, for example from the PAGES PALSEA activity would be worth considering, and citing in an explanation of why these were not part of the current study.

Thank you for this comment. We have added proglacial and modern lakes to our mapping and areal analyses, since these lakes were still widespread in portions of the study during the early Holocene. We also cited recent reconstructions of sea level change (from PALSEA) while noting that for the Holocene time period and continental-scale study presented here, these sea level effects are not expected to have a major effect on our reconstructions.

In the interpolated maps, the parts of the study domain that show no data I assume are because the "confidence region" (CR) was greater than the threshold of 9, for example in much of Mexico in the early Holocene. It would be helpful to see the CR maps themselves included among the supplementary figures. Looking at Figure 1, there are only 3 or maybe 4 sites in all of Mexico, so it is hard to understand, especially given the climatic and topographic diversity of Mexico, that there is much power in the interpolations over that space.

We agree that this is helpful, and have added maps of uncertainty for all time periods shown in the manuscript (Supp. Fig. S3; see also response to Reviewer 1).

We have set a fixed domain size that excludes grid cells that do not have reconstructions for all time periods. This excludes the Mexico region, as well as grid cells along the northernmost latitudes. Reconstructions are not temporally comprehensive for these regions due to a lack of records. Setting a fixed domain size removes confounding effects that may arise from the changing number of included grid cells.

All of the data products presented here (point-based and gridded maps) must be freely released on zenodo.org or other open-access data repository that provides a DOI upon final publication of the paper. The gridded maps should be provided in the earth system modeling-standard netCDF format.

Reviewer 1 raised the same point. We have archived the North American Holocene land cover reconstructions from both REVEALS and REVEALS-GMRF on Dryad: <a href="https://doi.org/10.5061/dryad.c2fgz61m5">https://doi.org/10.5061/dryad.c2fgz61m5</a>.

**A few notes on presentation**

As "land use" is generally accepted to be an activity that is unique to humans, it is not necessary to qualify the term with "human land use" or "anthropogenic land use" in the manuscript. In the interest of conciseness, please just use "land use" alone throughout the manuscript, or maybe define it once at the beginning of the text.

This is a good point, and we have removed most of these usages. We kept 'human land use' in a few locations where we felt it was important to emphasize the human component.

I found the constant switching back and forth between scientific names and common names for taxa distracting and sometimes confusing. Use of both nomenclatures even occurs in a single sentence (e.g., lines 435-436). I ask the authors to pick one nomenclature system and stick with it throughout the entire manuscript.

We have made this change, sticking with scientific names throughout.

Please use a thinner line thickness in all of the maps presented in the manuscript and supplement. The heavy line weight around the ice sheets and coastline distracts from the content. Perhaps the ice sheets could be plotted in a blue or brighter, contrasting color as polygons, without any outline at all.

We have revised the ice and coastline colors accordingly.

**Specific comments**

Lines 48-49

It is not at all clear how changes in the abundance of hemlock could have been significant enough to have a biogeophysical feedback to climate; see further comments below. We address this point below.

Lines 168-169

Please explain briefly how relative abundances are calculated when some taxa are ignored? Is there an "all other taxa" bin? Or are only abundances relative to the considered taxa included? What happens when a taxon that is considered to be important in terms of land cover, even locally, is not part of those used in the REVEALS model?

We have addressed this in our revisions to the Methods section. The standard REVEALS workflow does not include an "all other taxa" bin. This is because of the variability in PPEs and fall speeds among taxa that would be included in such a bin. In this work, we first translated the Neotoma taxonomy to the Whitmore taxonomy (ref); this resulted in a list of 47 taxa. We identified the taxa that were most abundant and indicators of land cover type, of which there

were 33. There were corresponding PPE and fallspeed values for all of these taxa. The set excluded results in a total of about 0.5% of the total pollen grains counted (for North American Holocene). We have added some text to clarify these decisions, and included a list of the taxa that were excluded in the supplement.

**Line 235**

Approximately how does the grid resolution of the 1x1 degree interpolated surface compare to the 10,000km2 area represented by a REVEALS reconstruction noted on line 179? Naturally it changes by latitude, but it would be helpful to put a comparative statement here.

We have added text to indicate the approximate grid cell size and reason for this decision.

**Lines 254-255**

Here where CR is introduced, it would be good to call out supplementary figures here showing this value in map form for all periods.

As indicated above and in our response to Reviewer 1, we have added maps of reconstruction uncertainty to the supplementary information (Supp. Fig. S3).

**Lines 260-261**

I understand that the LandCover6k grid was specified as 1x1 geographic degrees, but wouldn't it have made more sense to do the original work on an equal-area grid and then only reproject the data in a final step? At the very least it would have made interpretation of the maps more straightforward, and would be similar to earlier work (Williams, 2003; Williams et al., 2004).

Our primary goal was to maintain consistency with the other LandCover6k papers. We are publicly sharing the results (as NetCDF files) in 1x1 degrees, but in our data visualizations, we've reprojected maps to Albers equal area, using the same projection parameters as in the earlier papers by Williams et al.

**Figure 1**

Could you plot the 1x1 degree graticule on this map using a very thin line in an unobtrusive color? It would make interpretation of the grid resolution of the other maps easier. We agree this would be useful. We experimented with adding this graticule, but it obscured figure legibility so we did not include this.

**Figure 1**

Given the very high density of sites, it seems strange that nearly all of Minnesota is not included in any of the regional boxes. The choice to exclude this area deserves some explanation.

Our selection of regions was not intended to be comprehensive (this would have made for a very long and dull paper). Rather, we picked selected regions, based on a) whether we saw an interesting trend in land cover to which we wanted to call the reader's attention and/or b) describing changes in the west, which has been less intensively studied. Other regions certainly could have been chosen, and Minnesota definitely has its merits. We have expanded a brief note about these criteria to the manuscript, in the last paragraph of the methods.

**Figure 2**

Use a thinner line weight, or no line at all for the ice sheet outline (as noted above). We have revised the map color scheme accordingly.

**Figure 2**

To aid in quickly interpreting the plots and to provide better consistency with the rest of the figures, please plot the land cover fractional surfaces in the same colors as used in Fig. 3 and the other timeseries plots. That is to say, plot the first column of maps in shades of green, the second in shades of blue, and the third in shades of orange.

We recognize that this change would make it easier to link the time series figures with the land cover maps, but one of the primary design objectives of the map series is to facilitate comparisons of land cover across the maps. Using different colors in land cover maps would make this difficult, and would require three legends instead of one. Given this, we have opted to keep the single color scheme for the maps.

**Figure 2**

As noted in the supplement the three interpolated surfaces sum across to 100% in each row, and there is no "missing" fraction that represents bare ground. In the Arctic and in desert areas, the landscape is not 100% vegetated. This information should not be buried in the supplement, and needs to be clearly noted when the main figures are presented in the figure caption and body text. It should further be noted as a limitation and explained why this is not the case in the main manuscript text.

Agreed, we have clarified this in the main text by adding a sentence to the end of Section 2.2: "Note that bare ground cannot be detected by pollen-based land cover mapping and so is not included as a potential land cover type; presumably this missing land cover type is usually misclassified as OVL." and a similar sentence to the end of Section 3.3.4.

**Line 311**

Given that there are only 3 sites in Mexico, is the spatial domain of the study justified? Wouldn't a maximum distance buffer around nearest site be better - e.g., up to 100 km apart (corresponding to the REVEALS indicative catchment area)? As noted above there is a distance filter on the grid based on the CR value, but it would be interesting to see how this translates into distance from a site. Some statistics, such as the max distance from any site in the interpolation, would be helpful, even if only in the supplementary materials.

We have established a standard spatial domain, reducing the size of the study domain to omit grid cells for which there are reconstructions only in a few time periods (e.g. Mexico and northernmost latitudinal grid cells). We agree that a quantitative assessment relating sample density to reconstruction uncertainty is useful, and are working on a subsequent manuscript to explore strategic site selection based on the relationship between sampling density and uncertainty. In the current framework, the GMRF post-hoc interpolator does not account for

pollen-vegetation processes. The spatial dependence is determined by the spatial dependence in the REVEALS reconstructions. As such, this means that in regions with few grid cells for which there are REVEALS reconstructions, the REVEALS-GMRF approach may miss local to regional variability in land cover composition.

**Line 320**

The number of gridcells contributing to the curves presented in Figure 3 changes based on ice area, apparently not sea level, but also CR value. Can we see an additional curve on this figure showing the total area in the spatial domain contributing to the cover estimate? By establishing a standard domain size (see responses above), we do not need to include a curve representing changes in domain size over time.

**Lines 435-436**

In this sentence, and others, please just choose one form of plant nomenclature or the other, and stick with it.

**Fixed.**

**Lines 541-543**

"... desert, steppe, and other open-land arid ecosystems are likely to be underrepresented in these reconstructions, due to a scarcity of dryland sites" yet the interpolated maps and timeseries curves imply continuous vegetation cover (without bare ground), if I understand correctly. This limitation of the methodology should be further described and justified.

As noted above, we have added two sentences to the ms. that make this point, to the end of Sections 2.2 and 3.3.4.

**Line 566**

Is there really nothing to say here about sea level dynamics over the period?

We have added to this paragraph a mention of sea level changes as another environmental driver.

**Line 625**

I suggest a small rewording of this sentence to: "During the late Holocene, the growth of Indigenous populations and intensification of land use in the Americas had increasing effects on land cover. Understanding the interactions among..."

**We have made this change.**

**Line 632**

Evidence for dense populations and land use in North America are dismissed here, yet a number of examples of this are provided in the following paragraph. This sentence could be reworded to better tie to what is coming next.

We reworded the opening two sentences of this paragraph to address this comment and also Reviewer 1's note that Section 4.1 needed to be more closely tied to the manuscript results. The new opener now reads "In the land cover reconstructions presented here, the increase in prevalence of open lands after 0.5 ka in North America and some subregions (Figs. 3, 4, 7) is attributable to land use, but the role of land use for earlier time periods remains unclear. Land use prior to EuroAmerican settlement clearly altered land cover at some sites in North America, but land use effects are not easily detected at the regional to continental scales addressed here."

**Paragraph starting on line 660**

What is the purpose of this paragraph? Can it be tied back to the data presented in the current study?

We added a sentence that ties this paragraph back to the results. The general goal of this paragraph is to at least briefly note the major effects of EuroAmerican land use, without going deep into this topic, as previous papers have covered this topic well.

Section starting on line 668 (4.1.1)

This section needs to be tied back more clearly to the findings in the current study, at least speculatively. The section reads like a review paper now and there is nothing new in here.

Per comments from Reviewer 1, we have kept this paragraph, while shortening its treatment of biogeophysical feedbacks and adding a paragraph on biogeochemical (carbon cycle) feedbacks.

Line 675-676

The full name of the "TEMPO" acronym could be removed here and just put in the bibliography. Fixed.

Paragraph starting on line 707

This paragraph does a very good job of explaining how the data synthesized in the current study ties back to previous work. It should be a model for how section 4.1.1 could be improved.

Thank you.

Line 715

"Great Plains"

Fixed.

Line 728-731

It is not clear from this section or from the maps or timeseries how large, in absolute terms, the coverage of T. Canadensis could have ever been. The paragraph seems to insinuate that it

could have been abundant enough to make a majority proportion of forest cover, therefore having a strong influence on, e.g., albedo. But... (see next comment)

See response to the next comment.

**Lines 731-732**

Am I missing something because I don't see a shift in the dominance in Fig. 4, which is always more than 50% summergreen trees and shrubs, with evergreen less than 30% cover fraction throughout the Holocene. Are you arguing that ETS forests were conifer-dominated? Otherwise, the albedo changes would have been very subtle, especially since T. Canadensis can persist in the understory for a century or longer and so while it is there and producing pollen, it will have no influence on summer albedo and relatively little on winter.

Yes, please note the interesting difference between Figure 4b and 4a. The time series in Figure 4b show little change, as Reviewer 2 notes, but they are averaging across a broad area. Figure 4a shows that there is a very large effect associated with the Tsuga collapse, with 40% changes in evergreen cover, but that these changes are concentrated in the eastern part of the study domain. Tsuga is a late-successional shade-tolerant tree and tends to be a canopy dominant in areas of low disturbance. We stand by our inference that this single-species collapse could have had a major effect on land-atmosphere interactions at regional to subregional scales, and perhaps more broadly, depending on how the teleconnections played out.

More broadly, this topic is a good example of how different phenomena are operating at different scales - one of the main points of this paper.

**Lines 741-743**

Looking at the summary figures, these changes must have all be very subtle. If not, then some further quantitative information should be highlighted here.

Based on the summary figures, the changes represented here are about 5% at a continental scale, which is worth reporting and discussing, because of the large spatial extent involved. (A global mean temperature increase of 2C is a big deal; a 2C increase locally not as much.) This is another topic that is a good example of how different effects manifest at different scales. The Tsuga collapse was very large but at subregional to regional scales; this is a smaller change but across a much larger spatial domain.

**Line 747**

Broadleaf summergreen trees have greater maximum evapotranspiration rates than needleleaf evergreens and this effect should also be mentioned here as it is probably more important than the summertime albedo differences.

We have reviewed the literature, and have found that there is no consensus about the relative evapotranspiration rates of different forest types, and that these rates are quite variable based on stand age, structure, and meteorological conditions. Given this we have added text.

**Line 753**

I am not convinced that there is anything more than "relatively subtle shifts in the proportions of summergreen and evergreen trees and shrubs" shown in the data presented here.

**See response above, for comment on L741-743.**

**Line 775-776**

If "... REVEALS estimates are sensitive to parameter choices..." then why didn't you not just explore a larger parameter space and make a range of reconstructions? Instead of just one? Seems like it would be an easy change and could lead to the preparation of a range of maps or uncertainty fields.

In this manuscript, we are focusing on the REVEALS protocol and a careful review of the resulting reconstructions. We believe that adding a sensitivity analysis of REVEALS parameterizations is beyond the scope of this paper.

**Line 784-785**

The sentence mentions that "...this approach does not mechanistically represent the underlying processes that link pollen to vegetation". The GMRF method also does not account for soil, slope, aspect, and other edaphic controls on vegetation cover. This should be mentioned.

We have added a sentence that indicates this.

**Line 828-830**

Here it is admitted that the changes in "... continental-scale fractional forest cover were broadly stable." This statement does not seem to support the idea that biogeophysical feedbacks between land and atmosphere would have been very important, in contrast to what is insinuated earlier in the manuscript. Some further explanation would be helpful here.

As noted above, different effects manifest at different scales, and this manuscript is designed to report phenomena across scales. During revisions, we reviewed and revised statements and sections to ensure that they are clearly associated with the appropriate scale of inference.

**References**

Williams, J. W. (2003). Variations in tree cover in North America since the last glacial maximum. Global and Planetary Change, 35(1-2), 1-23. doi:10.1016/S0921-8181(02)00088-7 Williams, J. W., Shuman, B. N., Webb, T., Bartlein, P. J., & Leduc, P. L. (2004). Late-Quaternary Vegetation Dynamics in North America: Scaling from Taxa to Biomes. Ecological Monographs, 74(2), 309-334. doi:10.1890/02-4045